# Mitigate the Gap: Improving Cross-Modal Alignment in CLIP

**Sedigheh Eslami**
Hasso Plattner Institute/University of Potsdam
Potsdam, Germany
sedigheh.eslami@hpi.de

**Gerard de Melo**
Hasso Plattner Institute/University of Potsdam
Potsdam, Germany
gerard.demelo@hpi.de

## Abstract

Contrastive Language–Image Pre-training (CLIP) has manifested remarkable improvements in zero-shot classification and cross-modal vision-language tasks. Yet, from a geometrical point of view, the CLIP embedding space has been found to have a pronounced modality gap. This gap renders the embedding space overly sparse and disconnected, with different modalities being densely distributed in distinct subregions of the hypersphere. In this work, we propose AlignCLIP, in order to improve the alignment between text and image embeddings, and thereby reduce the modality gap. AlignCLIP increases the cross-modal alignment, and yields gains across several zero-shot and fine-tuning downstream evaluations by sharing the learnable parameters between the modality encoders and a semantically-regularized separation objective function on the uni-modal embeddings. The source code and model checkpoints for reproducing our experiments are available at https://github.com/sarahESL/AlignCLIP.

## 1 Introduction

One of the most prominent vision–language models is OpenAI's Contrastive Language–Image Pre-training (CLIP) model (Radford et al., 2021). CLIP is a dual-stream vision–language encoder trained for learning a joint representation space, in which image and text modalities can be jointly embedded. It has demonstrated exceptional zero-shot capabilities for image classification, multi-modal retrieval as well as robustness to natural distribution shifts, and is widely used in diverse domains (Eslami et al., 2023; Bleidt et al., 2024; Lin et al., 2022).

Despite the outstanding performance of CLIP, recent work has shed light on a pronounced *modality gap* in the CLIP embedding space (Liang et al., 2022; Tyshchuk et al., 2023; Schrodi et al., 2024), leading to large distances between image and text embeddings. We illustrate this phenomenon in Figure 1, which shows the DOSNES (Lu et al., 2019) projection of the CLIP-encoded image–text pairs from CC3M (Sharma et al., 2018) using the pre-trained ViT-B-32 backend. As can be seen, each modality densely populates a separate small subregion of CLIP's representation embedding space.

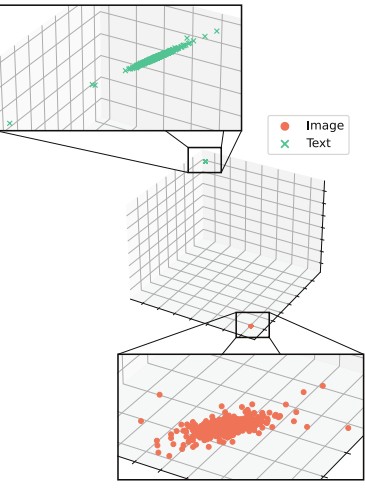

Figure 1: Modality gap in CLIP

Previous work (Liang et al., 2022; Schrodi et al., 2024; Tyshchuk et al., 2023) has studied the effects of closing the gap by translating the embeddings of one of the modalities to those of the other modality, using the average distance of the image–text pairs, e.g., shifting the image embedding $v_i$ for each image $i$ by $\Delta$ towards the language embedding $t_i$, in which $\Delta = \frac{1}{N}\sum_{i=1}^{N} t_i - \frac{1}{N}\sum_{i=1}^{N} v_i$ and $N$ is the total number of image–text pairs. However, such naive translations likely harm the relative distances of the *unpaired* images and texts, and therefore distort the meaningful structure of the embedding space. We show this by a simple 2D example in

Section A.1. In contrast, our work proposes to reduce the gap by first sharing the parameter space between the vision and language encoders and then imposing an intra-modality separation objective. The goal of sharing parameters between the encoders is to increase the inductive bias towards shared cross-modal encodings, and thereby, reducing the cross-modal gap. The intra-modality separation more directly aims at encouraging the embeddings within the visual modality to be reasonably expanded and pushed towards the language modality. Hence, each modality is pushed towards the embeddings of the opposite modality and thereby, the modality gap can get reduced. Our extensive experimental results support that the two aforementioned refinements show substantial improvement of the cross-modal alignment while improving the performance across a wide variety of downstream tasks.

To summarize, our contributions are as follows:

1. In Section A.1, we show that the previous naive approaches for closing the gap leads to damaged relative distances of the unpaired samples, which hurts the performance in downstream tasks.

2. Then, we propose sharing the learnable parameters of the vision and language encoders and study its effects on modality gap reduction and furthermore, on downstream performance.

3. In addition, we propose AlignCLIP, a semantically-regularized intra-modality separation objective function that enforces larger distances for the image embeddings of semantically dissimilar images, and at the same time, smaller distances on those of similar images.

4. We demonstrate that AlignCLIP achieves significant alignment of the modalities, and also substantially improves zero-shot classification, robustness to distribution shifts, and classification with linear probing.

5. We further investigate the impacts of AlignCLIP for the zero-shot and fine-tuned multi-modal retrieval tasks and provide a qualitative analysis of the image rankings as well as cosine similarities of the embeddings.

## 2 BACKGROUND, NOTATIONS AND CONCEPTS

Given a set of $N$ image–text pairs, we consider the CLIP image encoder $f_V$ to obtain the $l_2$-normalized vector $\vec{e}_v^i \in \mathbb{R}^d$ as a $d$-dimensional embedding vector for image $v_i$, i.e., $f_V(v_i) = \vec{e}_v^i$, and the CLIP text encoder $f_T$ to obtain the $l_2$-normalized text embedding $\vec{e}_t^i \in \mathbb{R}^d$ for the text sample $t_i$, i.e., $f_T(t_i) = \vec{e}_t^i$. We denote a batch of encoded image–text pairs by $E_v \in \mathbb{R}^{b \times d}$ for images and $E_t \in \mathbb{R}^{b \times d}$ for texts, where $b$ refers to the batch size.

**Background on CLIP**. The contrastive objective in CLIP is the average of the vision to language and the language to vision Info-NCE contrastive loss functions formulated as:

$$\mathcal{L}_{v \to l} = \frac{-1}{N} \sum_{i=1}^{N} \log \frac{\exp[(\vec{e}_v^i \cdot \vec{e}_t^i) / \tau]}{\sum_{j=1}^{N} \exp[(\vec{e}_v^i \cdot \vec{e}_t^j) / \tau]}, \quad \mathcal{L}_{l \to v} = \frac{-1}{N} \sum_{j=1}^{N} \log \frac{\exp[(\vec{e}_v^j \cdot \vec{e}_t^j) / \tau]}{\sum_{i=1}^{N} \exp[(\vec{e}_v^i \cdot \vec{e}_t^j) / \tau]},$$
(1)

respectively, where $\tau$ is a learnable temperature parameter. The overall CLIP loss is then:

$$\mathcal{L}_{\text{clip}} = \frac{1}{2}[\mathcal{L}_{v \to l} + \mathcal{L}_{l \to v}].$$
(2)

In practice, a symmetric cross-entropy loss is employed using the vision and language logits. The label $y \in \mathbb{R}$, which is the index of the image–text pair in the batch, represents the correspondence of the paired samples. For a batch of image–text pairs, $Y \in \mathbb{R}^b$ denotes the set of labels. The visual and textual logits, $\hat{y}_v \in \mathbb{R}^{b \times b}$ and $\hat{y}_t \in \mathbb{R}^{b \times b}$, are then calculated as:

$$\hat{y}_v = \exp(\tau) E_v E_t^{\mathsf{T}} \quad , \quad \hat{y}_t = \hat{y}_v^{\mathsf{T}},$$
(3)

respectively. Ultimately, the overall CLIP loss is calculated using the cross-entropy loss, $H$, as:

$$\mathcal{L}_{\text{clip}} = \frac{1}{2}[H(\hat{y}_v, Y) + H(\hat{y}_t, Y)].$$
(4)

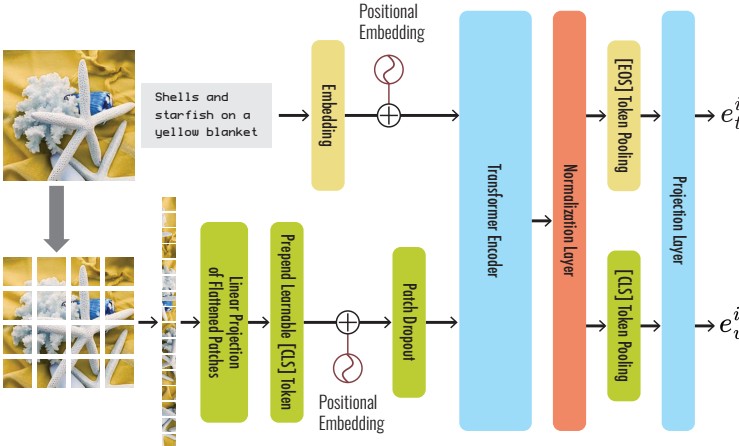

Figure 2: Overview of sharing the transformer and projection layer in SharedCLIP.

**Cross-Modal Alignment Score.** In contrastive representation learning, the goal is to learn similar representations for positive pairs and distant representations for irrelevant negative samples. In cross-modal vision–language learning, often the paired image–texts form the positive pair distribution. Alignment entails mapping positive pairs to close embedding vectors such that a perfect alignment is achieved when $f(x_1) = f(x_2)$ for a given encoding function $f$ and a randomly drawn positive pair of samples $x_1, x_2$ (Wang & Isola, 2020). In the multi-modal scenario, we are particularly interested in studying the alignment property for the CLIP embedding space, as it represents the modality gap (Jiang et al., 2023), and therefore, we define perfect alignment as $f_T(t_i) = f_V(v_i)$.

We adopt the alignment measurement proposed by Goel et al. (2022) and define it as the average cosine similarity between the paired image and text embeddings in the CLIP embedding space:

$$\text{alignment} = \frac{1}{N} \sum_{i=1}^{N} \vec{e}_v^i \cdot \vec{e}_t^i, \qquad \text{alignment} \in [-1, 1]. \tag{5}$$

Higher scores demonstrate better alignment, and, therefore, a decrease in the modality gap.

## 3 ALIGNCLIP

We attain cross-modal alignment in AlignCLIP by two main refinements: sharing the parameter space between the vision and language encoders, as well as learning to spread out and push apart the uni-modal embeddings. The following subsections describe each of these refinements in detail.

### 3.1 SHARING THE LEARNABLE PARAMETER SPACE IN CLIP

AlignCLIP employs a transformer-based encoder architecture (Vaswani et al., 2017), where the transformer is shared between the two vision and language modality encoders. We suspect that one of the main reasons that the modality gap exists in the original CLIP's embedding space is the fact that with a dual-stream encoder architecture, each modality has a separate disentangled set of parameters for optimization, i.e., $f_T$ has the parameters $\theta$ and $f_V$ has $\theta'$, which are initialized and optimized disjointly. As a result, even with the same encoder architecture for each modality, the final learned parameters $\theta$ and $\theta'$ can diverge quite radically, leading to distinct functions that map each modality to completely different subregions of the embedding space. In other words, training two transformers separately has no explicit inductive bias towards a shared region of the encoding space.

Therefore, we seek to align the outputs of the vision and language encoding functions by increasing the overlap of $\theta$ and $\theta'$ to the extent possible. We achieve this by sharing the layers of the vision and language encoding towers. Specifically, we share the transformer encoder as well as the projection layer between the vision and language modalities. Sharing the layers between the modality encoders increases the inductive bias towards a shared region of the embedding space, and therefore, reduces

the modality gap. We utilize a standard transformer encoder architecture (Dosovitskiy et al., 2020). An overview of our refined overall model architecture is given in Figure 2. Yellow components are designated for the language modality, green components for the visual modality, and the blue parts are shared between the two modalities. For encoding texts, the yellow and blue parts of the model actively contribute in the embedding calculations. Similarly, when encoding images, the green components as well as the blue ones are invoked for the computation. Following the original CLIP, we use the $[EOS]$ token pooling for text embeddings and the $[CLS]$ token embedding for the image embedding. We refer to this architecture as SharedCLIP throughout the rest of the paper. Sharing transformers in CLIP has been previously investigated from the perspective of downstream performance (You et al., 2022). In this work, we extend the sharing to the extent possible, i.e., we include the projection layer as well, and study the impacts with respect to the cross-modal alignment property.

### 3.2 INTRA-MODALITY SEPARATION

As shown in Figure 1, each modality resides in a distinct dense subregion of the CLIP embedding space. We hypothesize that this phenomenon is a direct result of the *cross-modal* contrastive objective in CLIP, which merely optimizes the relative distances of image embeddings and text embeddings. The cross-modal contrastive loss alone is not sufficient for imposing meaningful distances within the uni-modal embeddings, i.e., pairwise distances of text embeddings or pairwise distances of image embeddings. Therefore, we define an objective function that enforces reasonable distances within each modality by separating the uni-modal embeddings that are semantically dissimilar. In other words, we impose a semantically-regularized Intra-Modality Separation (IMSep) in addition to the CLIP's objective function.

IMSep is achieved by an InfoNCE contrastive loss, in which image–text pairs are the positive samples and all pairwise image–image combinations are considered as negative samples, respectively:

$$\mathcal{L} = \frac{-1}{N} \sum_{i=1}^{N} \log \frac{\exp[(\vec{e}_v^i \cdot \vec{e}_t^i) \, / \, \tau]}{\sum_{\substack{j=1, \\ i \neq j}}^{N} \exp[(\vec{e}_v^i \cdot \vec{e}_v^j) \, / \, \tau]}. \tag{6}$$

The distinguishing role of the IMSep loss, in comparison to the CLIP loss, is the enforcement of minimizing the pairwise image–image cosine similarities, i.e., the denominator in Eq. 6. As a result, this objective spreads apart different image embeddings. However, one should notice that while pushing the image embeddings away from each other, some samples might indeed bear some semantic similarity, and therefore ought not to be substantially separated. To account for this, we regularize the intra-modality separation with respect to the pairwise semantic distances of samples. Denoting the semantic distance of the image sample $i$ from $j$ by $\mathcal{D}_{ij}$, the regularized loss becomes:

$$\mathcal{L} = \frac{-1}{N} \sum_{i=1}^{N} \log \frac{\exp[(\vec{e}_v^i \cdot \vec{e}_t^i) \, / \, \tau]}{\sum_{\substack{j=1, \\ i \neq j}}^{N} \exp[(\mathcal{D}_{ij} \cdot \vec{e}_v^i \cdot \vec{e}_v^j) \, / \, \tau]}. \tag{7}$$

In Eq. 7, the distance $\mathcal{D}_{ij}$ controls the minimization of $(\vec{e}_v^i \cdot \vec{e}_v^j)$ by reducing these in-modality dot product values if the samples are semantically close (similar), and therefore, the denominator does not get strongly minimized, since the values are already small. Conversely, the denominator obtains larger values when $\mathcal{D}_{ij}$ is large, when images are semantically dissimilar, and therefore, the denominator gets strongly minimized.

In practice, IMSep aims at optimizing Eq. 7 using the cross-entropy loss. Given a batch of encoded image–text pairs, IMSep creates the $\hat{y}_{vsep}$ logits and then minimizes the cross-entropy loss over $Y$ and $\hat{y}_{vsep}$. To this end, first the pairwise cosine similarities of the images are calculated by:

$$\mathcal{V} = E_v E_v^\intercal, \quad \text{where} \quad \mathcal{V} \in \mathbb{R}^{b \times b}. \tag{8}$$

For the semantic regularization, and therefore, calculating $\mathcal{D}_{ij}$, we utilize the pairing texts of the images as the semantic supervision signal, and invoke a pre-trained sentence encoder for encoding each text as $\vec{e}_s^i \in \mathbb{R}^d$. We denote the corresponding batch of semantically encoded texts as $E_s \in \mathbb{R}^{b \times d}$, and proceed to calculate the distance matrix $D \in \mathbb{R}^{b \times b}$ from the similarity matrix $S \in \mathbb{R}^{b \times b}$:

$$\mathcal{S} = \frac{E_s E_s^\intercal}{\|E_s\|^2}, \qquad \mathcal{D} = 1 - \mathcal{S}, \tag{9}$$

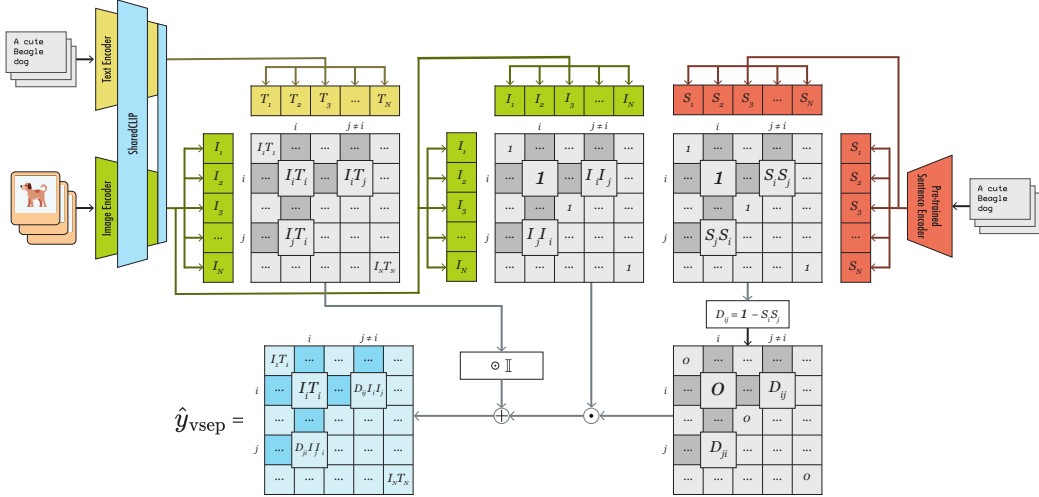

Figure 3: Schematic summary of the Intra-Modality Separation approach in AlignCLIP.

We rely on $\mathcal{D}$ in order to re-scale $\mathcal{V}$ by:

$$\mathcal{V}_\mathcal{D} = \mathcal{V} \odot \mathcal{D}, \quad \text{where} \quad \mathcal{V}_\mathcal{D} \in \mathbb{R}^{b \times b}, \tag{10}$$

and $\odot$ denotes the element-wise product. The re-scaling mechanism encourages reducing the dot product values in $\mathcal{V}$ for semantically similar images, while increasing it for dissimilar images.

Afterwards, for calculating the logits of positive samples, we calculate $\mathcal{M} = E_\mathrm{v} E_\mathrm{t}^\mathsf{T}$ and mask the non-diagonal values by $\mathrm{diag}(\mathcal{M}) = \mathbb{I} \odot \mathcal{M}$. We then obtain the $\hat{y}_\mathrm{vsep}$ logits by:

$$\hat{y}_\mathrm{vsep} = \exp(\tau) \cdot [\mathrm{diag}(\mathcal{M}) + \mathcal{V}_\mathcal{D}], \quad \hat{y}_\mathrm{vsep} \in \mathbb{R}^{b \times b}. \tag{11}$$

In Figure 3, our approach for obtaining IMSep is schematically summarized. Ultimately, we define the Intra-Modality Separation loss as:

$$\mathcal{L}_\mathrm{IMsep} = H(\hat{y}_\mathrm{vsep}, Y), \tag{12}$$

and adopt the core of the CLIP loss to represent the cross-modal separation:

$$\mathcal{L}_\mathrm{CRsep} = H(\hat{y}_\mathrm{v}, Y) + H(\hat{y}_\mathrm{t}, Y). \tag{13}$$

The final loss function optimized in AlignCLIP is:

$$\mathcal{L} = \mathcal{L}_\mathrm{CRsep} + \alpha \mathcal{L}_\mathrm{IMsep} \tag{14}$$

Note that it is sufficient to define the Intra-Modality Separation function for only one of the modalities, image in our case, since the cross-modal objective defined by $\mathcal{L}_\mathrm{CRsep}$ already behaves as a supervision for the other modality, text in our case, and enforces the Intra-Modality Separation in the opposite modality as well. To better support this statement, we extensively experiment on the effects of adding a similar Intra-Modality Separation on text embeddings in Section A.3 and show that it is sufficient to impose the Intra-Modality Separation on the image embeddings.

## 4 EXPERIMENTS

### 4.1 TRAINING DATASET AND SETUP

We used the Conceptual Caption 12M (CC12M) dataset (Changpinyo et al., 2021) for pre-training the models. In order to ensure a fair comparison of CLIP, we configured our setup for SharedCLIP and AlignCLIP to be similar to that of the original CLIP with the ViT-B-16 backend. The pre-trained semantic encoder utilized in AlignCLIP for re-scaling image–image cosine similarities is the SBERT all-mpnet-base-v2 model. Moreover, in terms of ablations, we train a version of AlignCLIP without sharing parameters, i.e., IMSep, and compare its performance. We trained all models from scratch using the OpenCLIP implementation (Cherti et al., 2023; Ilharco et al., 2021). In AlignCLIP, we set $\alpha = 0.5$. Section A.2 provides more details about the training setup.

| MODEL | CC3M | MSCOCO | IMAGENET-1K | CIFAR-100 | CIFAR-10 |
|-------|------|--------|-------------|-----------|----------|
| CLIP | 0.42 | 0.47 | 0.41 | 0.38 | 0.40 |
| SHAREDCLIP | 0.59 | 0.62 | 0.57 | 0.54 | 0.54 |
| IMSEP | 0.61 | 0.64 | 0.59 | 0.58 | 0.6 |
| ALIGNCLIP | **0.64** | **0.67** | **0.63** | **0.62** | **0.64** |

Table 1: Comparison of the alignment score.

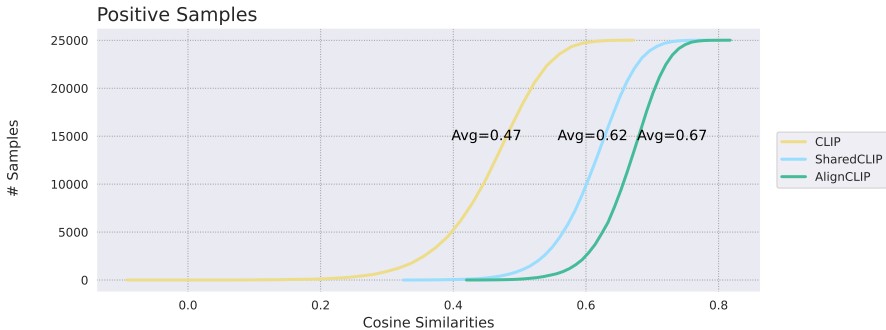

Figure 4: Cumulative distribution of pairwise cosine similarities of positive samples in MSCOCO.

## 4.2 CROSS-MODAL ALIGNMENT

We start by reporting and comparing the alignment scores when using CLIP, SharedCLIP, and AlignCLIP models on the validation sets from CC3M, MSCOCO as well as the ImageNet-1K, CIFAR-100, and CIFAR-10 test datasets. Table 1 summarizes the corresponding alignment scores. We observe that the original CLIP model has relatively low alignment scores, varying within $[0.38, 0.47]$, across all five datasets. In contrast, sharing the parameter space in SharedCLIP results in noticeable improvements of up to 0.17 in the alignment scores. We also observe that IMSep noticeably improves the alignment.Furthermore, using AlignCLIP yields even better alignment scores, ranging from 0.62 to 0.67. In other words, AlignCLIP reduces the average angle between the paired image–text embedding from about $70°$ to $47°$, which can be interpreted as reducing the modality gap.

We also plot the cumulative distribution of cosine similarities of the positive samples from MSCOCO when encoded using the original CLIP, SharedCLIP, and AlignCLIP in Figure 4. We find that using SharedCLIP noticeably shifts the distribution of the cosine similarity of positive samples towards higher similarity values. A higher similarity of positive samples, i.e., image–text pairs, means achieving greater cross-modal alignment, and thus a lower modality gap. Furthermore, AlignCLIP shifts the distribution even more to the right, resulting in a better reduction of the modality gap.

For a more comprehensive comparison of the distribution of each modality and studying the modality gap, we visualize the DOSNES projection of the encoded image–texts from the MSCOCO dataset in Figure 5. As can be seen, the uni-modal embeddings of the original CLIP are densely located on opposite sides of the hypersphere. In contrast, the embeddings get spread out, hence reducing the modality gap, when using the SharedCLIP model. This observation confirms our hypothesis that sharing layers in SharedCLIP adds greater inductive bias towards a shared embedding space, in comparison to training separate transformers in CLIP, and therefore, reduces the gap. Finally, the embedding space of AlignCLIP achieves the best spread of the uni-modal embeddings and substantially decreases the gap between image and text embeddings, showing the effectiveness of pushing the image embeddings apart from each other by the IMSep loss. In summary, Table 1, Figures 4 and 5 confirm that AlignCLIP achieves an embedding space with the least sparsity, least modality gap, and the best cross-modal alignment.

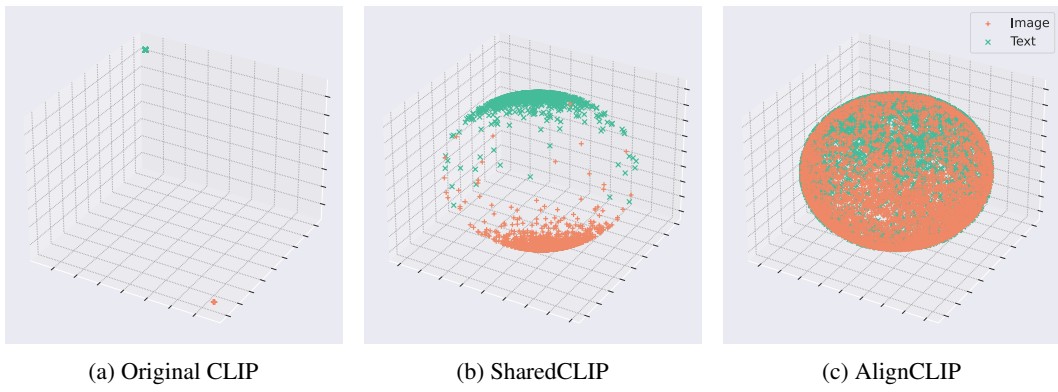

| (a) Original CLIP | (b) SharedCLIP | (c) AlignCLIP |

Figure 5: DOSNES visualization of the multi-modal embeddings using CC3M

| MODEL | IMAGENET-1K | | CIFAR-100 | | CIFAR-10 | | FLOWERS-102 | | STANFORD CARS | |
|---|---|---|---|---|---|---|---|---|---|---|
| | TOP1 | TOP5 | TOP1 | TOP5 | TOP1 | TOP5 | TOP1 | TOP5 | TOP1 | TOP5 |
| CLIP | 31.4 | 58.7 | 28.1 | 55.9 | 61.6 | 95.6 | 19.9 | 40.6 | 11.7 | 36.4 |
| SHAREDCLIP | 32.1 | 59.7 | 26.4 | 54.7 | 56.9 | 95.2 | 18.2 | 38.8 | 10.7 | 35.4 |
| IMSEP | 32 | 59 | 31 | 60.1 | 61.6 | 96.2 | **20.7** | **40.5** | 9.3 | 32.7 |
| ALIGNCLIP | **32.8** | **60.6** | **36.5** | **66.3** | **69.4** | **97.8** | 18.8 | 40.3 | **11.7** | **38.1** |

Table 2: Accuracy scores for zero-shot image classification.

## 4.3 CLASSIFICATION

CLIP's pre-training objective resulted in outstanding image classification performance when tested in either zero-shot or linear probing settings (Radford et al., 2021). Therefore, we further assess the performance of SharedCLIP and AlignCLIP in classification tasks.

**Zero-Shot Image Classification.** We conduct zero-shot classification experiments on ImageNet-1K (Russakovsky et al., 2015), CIFAR-100, CIFAR-10 (Krizhevsky et al., 2009), Flowers-102 (Nilsback & Zisserman, 2008), and Stanford Cars (Krause et al., 2013), with the combination of text prompts used by CLIP (Radford et al., 2021), e.g., "a photo of the {label}". The experimental results summarized in Table 2 show that SharedCLIP improves the classification performance on the ImageNet-1K dataset and reduces the accuracy on the rest of the datasets. In contrast, AlignCLIP yields the best scores across all five datasets. It achieves $1.4\%$ and $2\%$ improvement of Top-1 and Top-5 accuracy, respectively, on ImageNet-1K when compared to the original CLIP. On CIFAR-10, AlignCLIP achieves up to $8\%$ and $2\%$ improvements in Top-1 and Top-5 accuracy scores. Similarly, using AlignCLIP for CIFAR-100 results in about $8\%$ and $11\%$ enhancement in Top-1 and Top-5 accuracy scores. The trend of improvement is also observed on the Flowers-102 and Stanford Cars datasets. Our experiments thus evince that via sharing parameters and the additional IMSep separation, AlignCLIP enhances the downstream zero-shot image classification task.

**Linear Probing.** We further test the performance of SharedCLIP and AlignCLIP when performing linear probing for image classification and report the Top-1 accuracy results in Table 3. For all datasets, we train the linear classifier layer with a batch size of 128, for 30 epochs, with AdamW optimization, and a cosine scheduler with a starting learning rate of $5 \times 10^{-4}$. Table 3 shows that SharedCLIP generally outperforms CLIP in this setting and furthermore, AlignCLIP achieves the best scores, improving the image classification task with linear probing by up to 5% in comparison to CLIP. Overall, our experiments show that AlignCLIP and IMSep achieve the best classification performance in both zero-shot and linear probing settings.

## 4.4 ROBUSTNESS TO NATURAL DISTRIBUTION SHIFT

In the zero-shot image classification task, CLIP has additionally shown impressive robustness to natural distribution shifts and promising generalizability to out-of-distribution images. Therefore, we

| MODEL | IMAGENET-1K | CIFAR-100 | CIFAR-10 | FLOWERS-102 | STANFORD CARS |
|---|---|---|---|---|---|
| CLIP | 50.0 | 62.6 | 85.0 | 71.5 | 42.9 |
| SHAREDCLIP | 51.2 | 63.4 | 85.0 | 74.4 | 40.5 |
| IMSEP | **52.2** | 65.9 | 86.6 | **79.5** | **46.3** |
| ALIGNCLIP | 51.5 | **67.4** | **87.2** | 76.8 | 45.6 |

Table 3: Accuracy scores for image classification with linear probing.

| MODEL | IMAGENETV2 | | IMAGENET-R | | IMAGENET-A | | IMAGENETSKETCH | |
|---|---|---|---|---|---|---|---|---|
| | TOP1 | TOP5 | TOP1 | TOP5 | TOP1 | TOP5 | TOP1 | TOP5 |
| CLIP | 27.1 | 53.3 | 39.8 | 65.8 | 6.5 | 25.4 | 19.4 | 41.8 |
| SHAREDCLIP | 27.4 | 53.5 | 40.1 | 67.3 | 6.7 | **25.5** | 20.6 | **43.2** |
| IMSEP | 27.5 | 53.3 | 40.9 | **67.5** | 6.7 | 24.6 | 20.2 | 43.1 |
| ALIGNCLIP | **29.1** | **54.4** | **41.2** | 67.3 | **7.0** | 24.6 | **20.7** | **43.2** |

Table 4: Accuracy scores for zero-shot classification and natural distribution shift.

expand our evaluations and investigate to what extent SharedCLIP and AlignCLIP are affected by natural distribution shifts. We use the ImageNetV2, ImageNet-R, ImageNet-A, and ImageNetSketch datasets for these evaluations and report the corresponding results in Table 4, in terms of Top-1 and Top-5 accuracy. An initial observation is that SharedCLIP generally improves the classification accuracy in comparison to the CLIP model. Additionally, when comparing to the CLIP model, AlignCLIP achieves about $2\%$ improvement in Top-1 and Top-5 accuracy scores, leading to the conclusion that sharing parameters and applying IMSep in AlignCLIP improve the robustness to natural distribution shifts in comparison to the original CLIP model.

## 4.5 MULTI-MODAL RETRIEVAL

In addition to classification, we evaluate SharedCLIP and AlignCLIP in the applications of zero-shot and fine-tuned image-to-text and text-to-image retrieval using the MSCOCO (Lin et al., 2014) and Flickr30K (Plummer et al., 2015) datasets. In all settings, the text prompt "a photo of the {caption}" is used. When fine-tuning, the batch size was set to 128 and the AdamW optimizer with learning rate $5 \times 10^{-6}$ and a weight decay of 0.2 was used. For the fine-tuning experiments, each model was fine-tuned for 8 and 20 epochs on MSCOCO and Flickr, respectively.

The results of the zero-shot and fine-tuned evaluations are reported in Table 5. Our experiments show that both SharedCLIP and AlignCLIP improve the retrieval results measured by $R@\{1, 5, 10\}$ on both datasets when compared to the original CLIP model. In addition, AlignCLIP achieves the best overall results in comparison to SharedCLIP. When testing text-to-image retrieval in the zero-shot setting on the MSCOCO dataset, SharedCLIP outperforms AlignCLIP at $R@10$ by less than $1\%$. After fine-tuning, the recall difference is reduced to $0.5\%$. Similarly, for the image-to-text retrieval on the Flickr dataset, SharedCLIP outperforms AlignCLIP by about $1\%$ at $R@\{1, 5\}$ and fine-tuning makes the performances marginally different.

For a more comprehensive understanding of the differences in results in the retrieval task, Figure 6A provides an example text query for which SharedCLIP achieves a higher $R@5$ image retrieval in comparison to AlignCLIP. We provide the images at rank 1-3 and observe that all models retrieve images that are semantically relevant for the query text, but might not necessarily be the ground truth. In this example, such characteristics of the dataset can particularly affect the evaluation results of AlignCLIP, due to the semantic regularization in IMSep, as it will make the semantically similar images closer to each other and can retrieve semantically close samples that are not labeled as the ground truth. We suspect that this phenomenon is more noticeable in the retrieval scenario in comparison to image classification datasets, as in the classification tasks, classes are more semantically disjoint. In Figure 8, we further provide the ground truth texts for each image. Moreover, Figure 9 provides more of such, and Figure 10 illustrates text rankings given sample image queries.

| | MODEL | MSCOCO | | | | | | FLICKR30K | | | | | |
|---|---|---|---|---|---|---|---|---|---|---|---|---|---|
| | | I → T | | | T → I | | | I → T | | | T → I | | |
| | | R@1 | R@5 | R@10 | R@1 | R@5 | R@10 | R@1 | R@5 | R@10 | R@1 | R@5 | R@10 |
| ZS | CLIP | 31.4 | 57.0 | 68.6 | 20.5 | 44.1 | 55.9 | 53.2 | 80.5 | 88.6 | 39.9 | 69.0 | 78.5 |
| | SHAREDCLIP | 33.6 | 59.6 | 70.8 | 21.8 | 45.4 | 57.3 | 58.3 | 83.6 | 89.8 | 42.6 | 70.0 | 79.1 |
| | IMSEP | 33.7 | 60.8 | 71.5 | 21.5 | 45.1 | 56.9 | 56.8 | 82.8 | 89.2 | 42 | 69.9 | 79.4 |
| | ALIGNCLIP | 34.0 | 59.7 | 70.9 | 21.9 | 45.0 | 56.8 | 57.2 | 82.3 | 89.5 | 41.8 | 70.2 | 79.1 |
| FT | CLIP | 39.6 | 67.5 | 78.3 | 26.7 | 53.4 | 65.7 | 64.8 | 87.8 | 93.9 | 47.4 | 75.9 | 84.1 |
| | SHAREDCLIP | 40.7 | 69.2 | 79.7 | **27.9** | **55.0** | 66.7 | **66.5** | **89.1** | 94.1 | 48.9 | 76.4 | 84.3 |
| | IMSEP | 41 | **69.8** | 79.7 | 27.8 | 54.5 | **66.8** | 64.3 | 87.5 | 93.6 | 48.5 | 76.5 | **84.8** |
| | ALIGNCLIP | **41.7** | 69.3 | **80.1** | 27.9 | 54.5 | 66.2 | 66 | **89.1** | **94.5** | **49.3** | **76.7** | 84.4 |

Table 5: Zero-shot (ZS) and fine-tuned (FT) cross-modal retrieval summarized with R@{1, 5, 10}. Blue color represents the best performance for ZS. Numbers in bold represent the best scores.

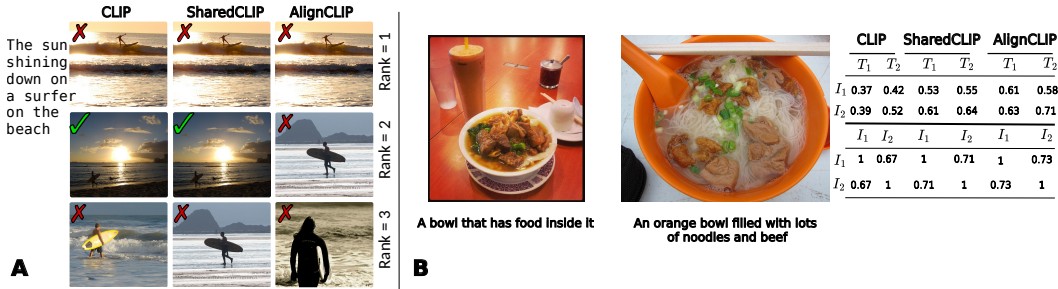

Figure 6: Qualitative analysis of the image rankings and cosine similarities on MSCOCO.

In contrary, in Figure 6B, we provide an example where AlignCLIP retrieves the most accurate caption or image. We use CLIP, SharedCLIP, and AlignCLIP to encode the images and texts, and report the cosine similarities on the right side. As can be seen, when querying the first image (on the left), the similarity of the first image and the second text (on the right) using the CLIP embeddings is higher in comparison to the ground truth caption, suggesting that the second text will, incorrectly, get selected as the predicted caption when using CLIP. This shortcoming still appears when using SharedCLIP for encoding the images and texts. However, when using AlignCLIP, the cosine similarity of the first image and the first text is higher in comparison to the second text, meaning that when querying the first image, the ground truth caption, which is semantically more correct in comparison to the second text, successfully gets selected. This suggests that the semantic regularization of the Intra-Modality Separation in AlignCLIP, which is calculated using the semantics of the text samples, potentially contributes to capturing more details in the captions, and, hence, improving the retrieval performance.

## 4.6 ABLATION STUDY

This section provides an ablation study on the effectiveness of the re-scaling mechanism proposed in Eq. 10. To this end, we compare the performance of AlignCLIP with and without the re-scaling mechanism on ImageNet-1K, CIFAR-100, CIFAR-10 for zero-shot classification, ImageNetV2 for robustness to distribution shift, and the MSCOCO and Flickr30K datasets for the retrieval task. For the retrieval tasks, R@10 is reported. From the summarized results in Table 6, we can substantially conclude that the re-scaling mechanism is effective in controlling the separation of similar image samples in the batch, and therefore, increasing the performance of the downstream tasks.

## 4.7 COMPARISON WITH STATE-OF-THE-ART MODELS

Table 7 provides a comparison of SharedCLIP and AlignCLIP with the state-of-the-art CLIP-based models. One of the prominent previous studies that also reports the alignment score is the CyCLIP model (Goel et al., 2022), which is trained using the CC3M dataset. For a fair comparison with CLIP

| MODEL | IMAGENET-1K | CIFAR-100 | CIFAR-10 | IMAGENETV2 | MSCOCO | | FLICKR | |
|---|---|---|---|---|---|---|---|---|
| | TOP1 | TOP1 | TOP1 | TOP1 | I → T | T → I | I → T | T → I |
| ALIGNCLIP-W/O RESCALING | 32.8 | 34.7 | 64.2 | 27.9 | **71.4** | 56.4 | 89.5 | 78 |
| ALIGNCLIP | 32.8 | **36.5** | **69.4** | **29.1** | 70.9 | **56.8** | 89.5 | **79.1** |

Table 6: Ablation study on the re-scaling mechanism in AlignCLIP.

| MODEL | IMAGENET-1K | | CIFAR-100 | | CIFAR-10 | | IMAGENET-A | | ALIGNMENT |
|---|---|---|---|---|---|---|---|---|---|
| | TOP1 | TOP5 | TOP1 | TOP5 | TOP1 | TOP5 | TOP1 | TOP5 | IMAGENET-1K |
| CLIP (RADFORD ET AL., 2021) | 16.5 | 34.1 | 23.7 | 49.7 | 51.3 | 92.7 | 4 | 15.8 | 0.36 |
| NAIVE SHIFTING (LIANG ET AL., 2022) | 16.6 | 34.6 | 24.5 | 52.5 | 51.2 | 91.2 | **43.3** | **68.8** | 0.16 (WITH LAMBDA=0.5) |
| CYCLIP (GOEL ET AL., 2022) | 15 | 32 | 21.8 | 48 | 44.7 | 85.4 | 3.3 | 13.6 | 0.32 |
| SHAREDCLIP | 17.4 | **34.9** | 26 | 52.1 | 57 | 92.9 | 3.7 | 14.5 | 0.45 |
| ALIGNCLIP | 17.6 | **34.9** | **26.3** | **54** | **57.9** | **93.7** | 4.2 | 16.1 | **0.63** |

Table 7: Comparison with the state-of-the-art CLIP-based models.

and CyCLIP, we re-train these models with the same hyperparameters as SharedCLIP and AlignCLIP. Furthermore, we train versions of SharedCLIP and AlignCLIP with the CC3M dataset and compare the results in Table 7. We also provide a comparison with the case of naive shifting proposed in Liang et al. (2022). Our results show a promising improvement of alignment scores while improving the downstream performance in comparison to the previous models.

# 5 RELATED WORK

**Modifications of CLIP.** Prior work has sought to improve several different aspects of CLIP, e.g., introducing a multi-positive contrastive loss in Lee et al. (2022), an additional image-based contrastive loss in Mu et al. (2022), optimizing the geometric consistency of the image and text embeddings in Goel et al. (2022), proposing a non-contrastive training regimen based on image–text pairs in Zhou et al. (2023), employing pairwise sigmoid loss for scaling up the batch size in Zhai et al. (2023), and domain adaptation by refining the fine-tuning in Hu et al. (2024). In contrast, AlignCLIP studies the cross-modal alignment in CLIP by investigating effective refinements of CLIP.

**Modality Gap in CLIP.** The modality gap in CLIP was first studied by Liang et al. (2022). They showed that the modality gap is caused by a combination of the model initialization and the contrastive loss optimization, and attempted to close the gap by shifting the embeddings. In a similar study, Tyshchuk et al. (2023) measured the alignment of image and text embeddings in CLIP with a focus on isotropic properties. More recently, Schrodi et al. (2024) showed that one of the key factors contributing to both the modality gap and the bias in CLIP is the information imbalance between the two modalities.Furthermore, Jiang et al. (2023) showed that exact modality alignment is suboptimal for downstream tasks, as the prediction error has a lower bound that is dependent on the modality information gap. Our AlignCLIP work shows that naive shifting of the embeddings distorts the relative distances of the unpaired samples, which can harm the downstream performance. We show that under the same information gap, i.e., using the same training datasets, parameter sharing between the modality encoders as well as semantically-regularized separation of the uni-modal embeddings reduce the gap while improving the downstream tasks.

# 6 CONCLUSION

This work presents AlignCLIP, a novel framework for enhancing the cross-modal alignment in the CLIP embedding space. AlignCLIP reduces the gap by increasing the inductive bias towards shared embeddings as well semantically-regularized modifications of the latent space. We show that AlignCLIP substantially improves the cross-modal alignment and further achieves the best results for image classification, robustness to distribution shifts and improves the multi-modal retrieval performance in comparison to CLIP. Our qualitative analysis on the retrieval task demonstrates the changes of the ranking of the neighborhoods for semantically similar images.

ACKNOWLEDGMENTS

We acknowledge the financial support from the German Federal Ministry for Education and Research (BMBF) within the project KI-Servicezentrum Berlin Brandenburg (01IS22092). We also thank Ali Ghaffaari, Maryam Hosseini, and Mina Rezaei for insightful discussions.

REPRODUCIBILITY STATEMENT

We have provided the details of our training and evaluations setups in Section A.2. Clear explanations of the shortcoming of the previous naive attempts at reducing the gap has been provided in Section A.1. Furthermore, the source code for training our models and model checkpoints for reproducing our are available at `https://github.com/sarahESL/AlignCLIP`.

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

## A    APPENDIX / SUPPLEMENTAL MATERIAL

### A.1    DESTRUCTIVE EFFECTS OF NAIVE EMBEDDINGS TRANSLATION FOR CLOSING THE GAP

In this section, we provide a 2D example for showing how the translation proposed in Liang et al. (2022) (and later adopted in Schrodi et al. (2024)) is not guaranteed to maintain the relative distances of the *unpaired* image–text samples, and therefore, can distort the meaningful structure of the embedding space. Figure 7 provides 2 image–text pairs, i.e., $(T_1, I_1)$ and $(T_2, I_2)$. As derived in Eq. 15, the distance of the first image is closest to the first text, and the distance of the second image is closest to the second text. However, in Eq. 16, after shifting by $\Delta$, as proposed by Liang et al. (2022), the distances of the second text to either of the images become equal and fail to maintain the relative distances after the shift. As a result, in downstream tasks, e.g., image retrieval given $T_2$, both $I_1$ and $I_2$ will be retrieved, which is incorrect.

$$
\begin{aligned}
d_{1,1} &= \|T_1 - I_1\| = \|[-2, -1]\| = \sqrt{4+1} = \sqrt{5} \\
d_{1,2} &= \|T_1 - I_2\| = \|[-2.5, 0.5]\| = \sqrt{6.25 + 0.25} = \sqrt{6.5} \\
d_{2,1} &= \|T_2 - I_1\| = \|[-1.5, -1.5]\| = \sqrt{2.25 + 2.25} = \sqrt{4.5} \\
d_{2,2} &= \|T_2 - I_2\| = \|[-2, 0]\| = \sqrt{4+0} = \sqrt{4} \\
& \qquad d_{1,1} < d_{1,2}, \quad d_{2,2} < d_{2,1}
\end{aligned}
\tag{15}
$$

$$
\begin{aligned}
\Delta &= (T_1 + T_2) - (I_1 + I_2) = [-1.5, 1.5] - [2.5, 2.5] = [-4, -1.5] \\
I_1' &= I_1 + \Delta = [-3, 0.5] \quad I_2' = I_2 + \Delta = [-2.5, -1] \\
d_{1,1}' &= \|T_1 - I_1'\| = \|[2, 0.5]\| = \sqrt{4 + 0.25} = \sqrt{4.25} \\
d_{1,2}' &= \|T_1 - I_2'\| = \|[1.5, 2]\| = \sqrt{2.25 + 4} = \sqrt{6.25} \\
d_{2,1}' &= \|T_2 - I_1'\| = \|[2.5, 0]\| = \sqrt{6.25} \\
d_{2,2}' &= \|T_2 - I_2'\| = \|[2, 1.5]\| = \sqrt{4 + 2.25} = \sqrt{6.25} \\
& \qquad d_{1,1}' < d_{1,2}', \quad d_{2,1}' = d_{2,2}'
\end{aligned}
\tag{16}
$$

### A.2    TRAINING DATASET AND SETUP

For pre-training, we used the CC12M dataset (Changpinyo et al., 2021). We adopted a transformer encoder consisting of 12 layers and 12 heads in CLIP, SharedCLIP, and AlignCLIP. The same input pre-processing and augmentations employed in CLIP have been used for SharedCLIP as well as AlignCLIP, including random cropping of images to the size $224 \times 224$. The image patch size for encoding visual data is set to $16 \times 16$. When encoding texts, the maximum sequence length is set to 77 tokens and the vocabulary size for the embedding layer is set to 49,408. The output embedding dimensionality for both the vision and language modalities is set to 768. We chose our setup to be quite similar to that of the original CLIP with the ViT-B-16 backend, in order to ensure a fair comparison of CLIP and our proposed model. The SBERT all-mpnet-base-v2 model has been utilized in AlignCLIP for re-scaling image–image cosine similarities in Eq. 10.

In order to fairly compare the effectiveness of each model, we trained all of the models from scratch using the CC12M dataset and the OpenCLIP implementation (Cherti et al., 2023; Ilharco et al., 2021). For all models, we used AdamW optimization with a starting learning rate of $1 \times 10^{-3}$, cosine scheduler, 10,000 warmup steps, and a weight decay of $0.1$. The initial temperature value for all models were set to 0.07. Each model was trained using an NVIDIA H100 GPU with batch size 512 for 30 epochs. We used the checkpoint from the last epoch in our evaluations of downstream experiments. In AlignCLIP, we set $\alpha = 0.5$.

### A.3    EFFECTS OF TEXT EMBEDDING SEPARATION

In Section 3.2, we proposed IMSep to enforce Intra-Modality Separation among the image embeddings. An alternative scenario could be to enforce the separation amongst the text embeddings, or even, to enforce the separation inside the image embeddings as well as the text embeddings. To examine these alternatives, we trained versions of AlignCLIP without image–image separation, instead

introducing a text–text separation, which we refer to as AlignCLIP-TT. We also used the re-scaling mechanism described in Eq. 10. Furthermore, we trained a version including both image-image embeddings separation and text-text embeddings separation, i.e., AlignCLIP-II-TT. We followed the same experimental setup as described in Section A.2 and compare the results for zero-shot image classification as well as linear probing in Tables 8 and 9, respectively. As can be seen, imposing the Intra-Modality Separation merely on the image embeddings yields the best results.

| MODEL | IMAGENET-1K | | CIFAR-100 | | CIFAR-10 | | FLOWERS-102 | | STANFORD CARS | |
|---|---|---|---|---|---|---|---|---|---|---|
| | TOP1 | TOP5 | TOP1 | TOP5 | TOP1 | TOP5 | TOP1 | TOP5 | TOP1 | TOP5 |
| ALIGNCLIP (TT) | 31.1 | 58.6 | 31.5 | 60.9 | 64.8 | 95.5 | 18.7 | 39.5 | 10.0 | 36.4 |
| ALIGNCLIP (II) | **32.8** | **60.6** | **36.5** | **66.4** | **69.3** | **97.8** | **18.8** | **40.3** | **11.8** | **38.1** |
| ALIGNCLIP (II-TT) | 32.4 | 60.0 | 31.2 | 61.8 | 66.4 | 96.9 | 18.7 | 39.9 | 11.8 | 37.1 |

Table 8: Accuracy scores for zero-shot image classification.

| MODEL | IMAGENET-1K | CIFAR-100 | CIFAR-10 | FLOWERS-102 | STANFORD CARS |
|---|---|---|---|---|---|
| ALIGNCLIP (TT) | 51.2 | 64.5 | 86.3 | 74.3 | 41.6 |
| ALIGNCLIP (II) | **51.5** | **67.4** | **87.2** | **76.8** | **45.6** |
| ALIGNCLIP (II-TT) | 50.3 | 63.4 | 85.8 | 72.2 | 42.2 |

Table 9: Accuracy scores for image classification with linear probing.

## A.4 QUALITATIVE EXAMPLES

In this section, further qualitative examples are provided. Figure 8 provides the ground truth text for the Top-3 images retrieved in Figure 6. Figure 9 provides further examples of the rankings when querying texts. Moreover, Figure 10 illustrates the text rankings when an image is queried.

## A.5 TREND OF THE TEMPERATURE VALUE

In this section, we provide an analysis of the learned temperature value in each of the CLIP, Shared-CLIP, and AlignCLIP models. We traced the value of $\frac{1}{\tau}$ and provide the plots in Figure 11.

| MODEL | CC3M | MSCOCO | IMAGENET-1K | CIFAR-100 | CIFAR-10 |
|---|---|---|---|---|---|
| CLIP | 0.4 | 0.44 | 0.38 | 0.58 | 0.36 |
| SHAREDCLIP | 0.58 | 0.61 | 0.57 | 0.56 | 0.57 |
| ALIGNCLIP | **0.67** | **0.69** | **0.66** | **0.64** | **0.65** |

Table 10: Comparison of the alignment score with ViT-S-16.

## A.6 EXPERIMENTAL RESULTS USING THE VIT-S-16 TRANSFORMER

In this section, we provide the results of using AlignCLIP with the ViT-S-16 backend in Tables 10–13.

| MODEL | IMAGENET-1K | | CIFAR-100 | | CIFAR-10 | | FLOWERS-102 | | STANFORD CARS | |
|---|---|---|---|---|---|---|---|---|---|---|
| | TOP1 | TOP5 | TOP1 | TOP5 | TOP1 | TOP5 | TOP1 | TOP5 | TOP1 | TOP5 |
| CLIP | 30.5 | 57.7 | 25.3 | 551. | 57.5 | 94.9 | 16.8 | 37.6 | 9.2 | 32 |
| SHAREDCLIP | 29.8 | 56.5 | 27.6 | 56.7 | 61.3 | 95.8 | 17.7 | 37 | 8.8 | 30.9 |
| ALIGNCLIP | **30.8** | **57.8** | **27.8** | **57.5** | **63.2** | **95.9** | **20.6** | **38.8** | **9.5** | **32.9** |

Table 11: Accuracy scores for zero-shot image classification with ViT-S-16.

| MODEL | IMAGENETV2 | | IMAGENET-R | | IMAGENET-A | | IMAGENETSKETCH | |
|---|---|---|---|---|---|---|---|---|
| | TOP1 | TOP5 | TOP1 | TOP5 | TOP1 | TOP5 | TOP1 | TOP5 |
| CLIP | 26.1 | 52 | 36.3 | 63 | 5.7 | 22.4 | 18 | **39.6** |
| SHAREDCLIP | 25.1 | 50.3 | 36.5 | 62.8 | 5.4 | 22.2 | 18 | **39.6** |
| ALIGNCLIP | **26.4** | **52.4** | **38.1** | **64.5** | **6.3** | **23.4** | **18.6** | **39.6** |

Table 12: Accuracy scores for zero-shot classification and natural distribution shift with ViT-S-16.

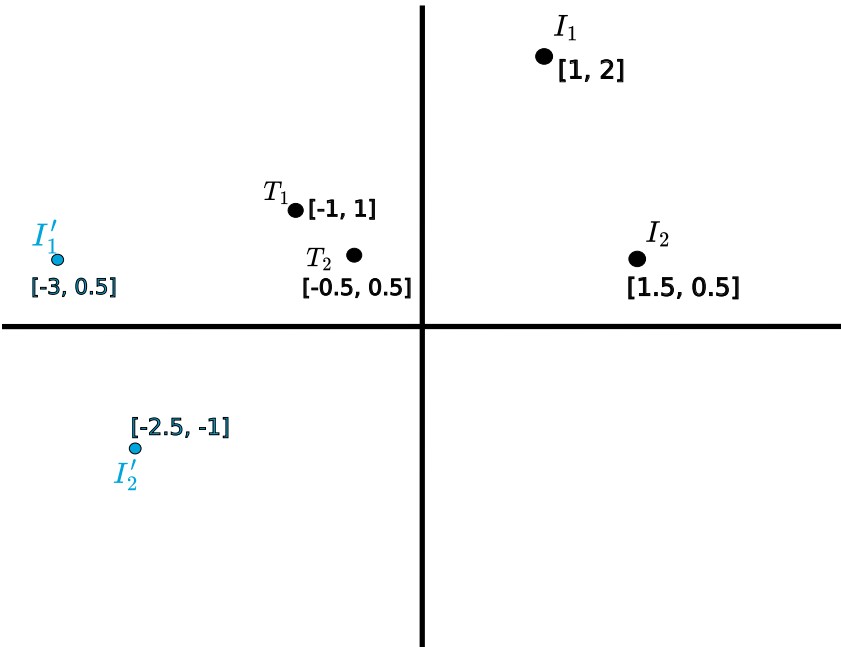

Figure 7: A 2D example illustrating that naive embedding shifts can distort the relative distance of the unpaired images and texts.

| | MODEL | MSCOCO | | | | | | FLICKR30K | | | | | |
|---|---|---|---|---|---|---|---|---|---|---|---|---|---|
| | | I → T | | | T → I | | | I → T | | | T → I | | |
| | | R@1 | R@5 | R@10 | R@1 | R@5 | R@10 | R@1 | R@5 | R@10 | R@1 | R@5 | R@10 |
| ZS | CLIP | 30.4 | 56.9 | 67.4 | 19.9 | 42 | 53.7 | 52.1 | 77.9 | 85.8 | 38.2 | 66 | 76.1 |
| | SHAREDCLIP | 31.3 | 57 | 68.1 | 20.3 | 42.2 | 54.1 | 52.1 | 80.9 | 87.4 | 38.6 | 66.4 | 76.1 |
| | ALIGNCLIP | 31.8 | 57.1 | 68.1 | 19.9 | 42.2 | 53.9 | 52.7 | 80.6 | 87.2 | 38.7 | 66.6 | 76.4 |
| FT | CLIP | 38.2 | 66 | 76.3 | 25 | 50.5 | 62.8 | 61 | 84.8 | 91.1 | 45.2 | 72.1 | 82 |
| | SHAREDCLIP | 38.5 | 66 | **76.8** | 25.2 | **51.1** | **63.4** | **61.1** | 85.1 | **91.7** | 45.4 | **73.4** | **82.2** |
| | ALIGNCLIP | **39** | **66.8** | 76.4 | **25.4** | 50.8 | 63 | **61.1** | **85.4** | 91.6 | **45.5** | 72.8 | 82 |

Table 13: Zero-shot (ZS) and fine-tuned (FT) cross-modal retrieval summarized with R@{1, 5, 10}. Blue color represents the best performance in the ZS setting. Numbers in bold represent the best scores with ViT-S-16.

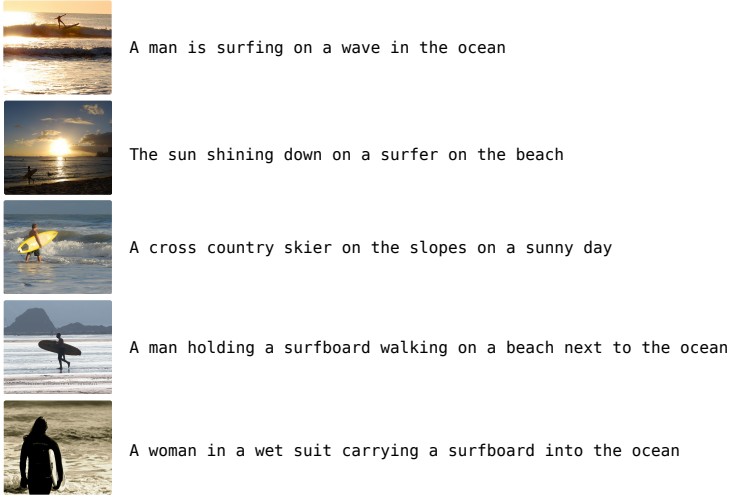

Figure 8: Ground truth texts for the Top-3 retrieved images in Figure 6.

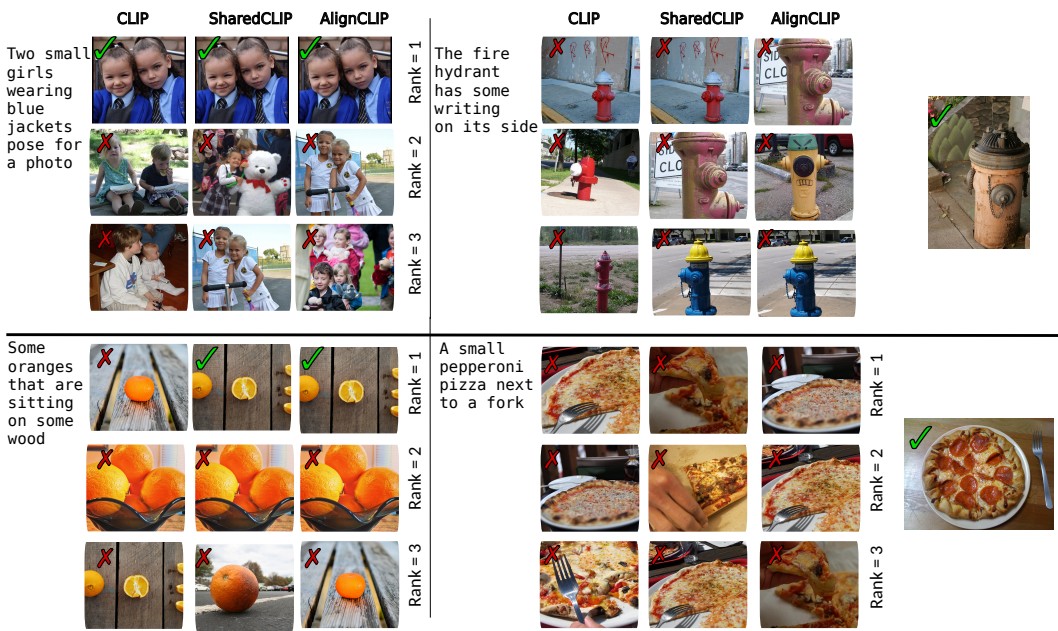

Figure 9: Qualitative analysis of the image rankings.

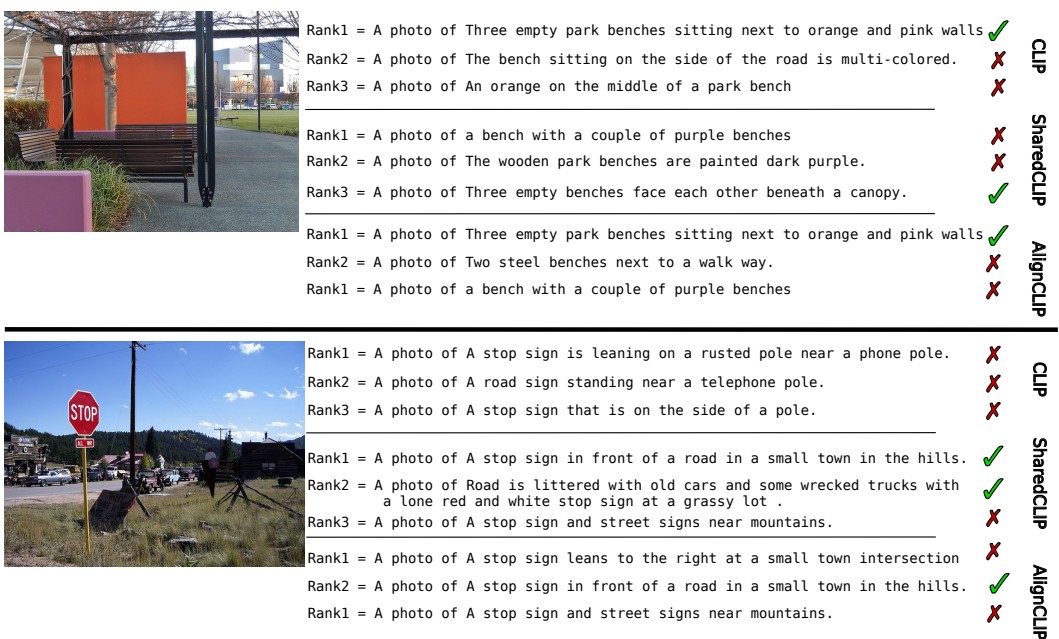

Figure 10: Qualitative analysis of the text rankings given image queries.

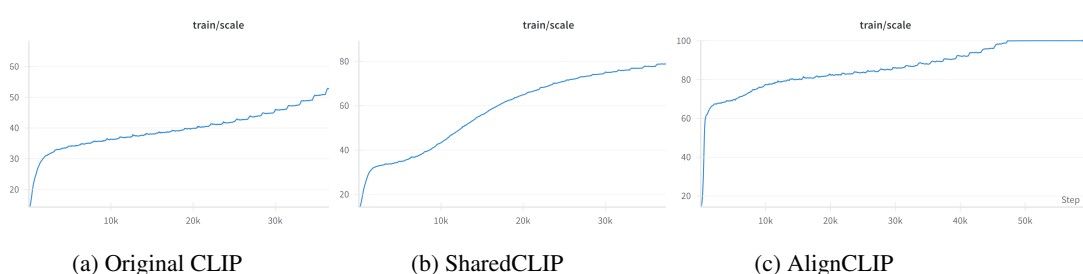

| (a) Original CLIP | (b) SharedCLIP | (c) AlignCLIP |
|---|---|---|

Figure 11: Trend of learnable temperature value: The Y-Axis shows the value of $\frac{1}{\tau}$ and the X-Axis shows the step number.

## B    LIMITATIONS

For the zero-shot classification experiments, we have tested the models using the ImageNet-1k, CIFAR-10, CIFAR-100, Stanford Cars, and Flower-102 datasets. Further classification analysis using benchmark datasets such as Hateful Memes has thus far not been conducted in this work. Moreover, this study is limited to the English language and should be followed up by further analyses on multilingual and cross-lingual representation learning. Additionally, the re-scaling mechanism in the intra-modality separation loss is dependant on the choice of pre-trained sentence encoder and has not been extensively benchmarked in this work. Pre-trained image encoders could as well be explored for providing the semantic supervision for the re-scaling mechanism.

## C    IMPACT STATEMENT

This paper presents research seeking to advance the field of machine learning at a fundamental level with regard to cross-modal representation learning. We acknowledge that this line of work has a broad range of potential societal implications. For instance, vision–language models may exhibit harmful biases and stereotypes, particularly when trained on data crawled from the Web. Due to their incorporation in prominent generative AI models, vision–language models may also contribute toward the model's ability to produce images portraying trademarked characters or notable figures. These sorts of concerns need to be carefully considered before incorporating such models into real-world applications.

