# OpenReview forum: "Mitigate the Gap: Improving Cross-Modal Alignment in CLIP"
_ICLR.cc/2025/Conference — ICLR 2025 Poster_

### Official Review · Reviewer_ZbcG · 2024-10-29

**Soundness:** 3
**Presentation:** 4
**Contribution:** 3
**Rating:** 6
**Confidence:** 3

**Summary:**

The paper proposed a novel method named AlignCLIP to address a recognized problem -- modality gap, for the CLIP network. The improved components include (i) a shared transformer and (ii) an intra-modality separation module. The proposed method surpassed the existing methods in most cases.

**Strengths:**

1. Technical solid paper with clear stated motivation and well organized content.
2. The proposed Intra-Modality Separation module is novel and is benefit to the community.
3. The proposed method surpassed the existing methods in most cases.

**Weaknesses:**

1. The idea of shared transformer is less novel. From my understanding, since the shared transformer for CLIP has been proposed by You et. al., then extending it to the projection layer is just an incremental improvements. The paper should discuss why the extension to the projection layer is necessary comparing to the existing methods. For example, in terms of "extend the sharing to the extent possible" in line #168m, what should we do if using CLIP-ResNet instead of CLIP-ViT.
2. Since the paper claims the (i) SharedCLIP (ii) IMSep module. It is not clear that which module contributes more on the overall performance. Since the SharedCLIP is strongly dependent on transformer structure, while IMSep module is more related to the general multi-modality contrastive learning task, therefore, the IMSep moduel has better expansibility for various model structures. Therefore, an ablation study on this is necessary, i.e. (original_CLIP+)SharedCLIP vs original_CLIP+IMSep, if I missed this experiment, please point it out during the rebuttal.
3. It's necessary to add some discussion for Table.5, regarding the COCO T->I. It seems the SharedCLIP works better in the T->I scenario, but what reason cause the degradation while adding IMSep? Some discussion and analysis regarding why this degradation occurs would be benifit for researchers to understand the limitation of the proposed model, for example, is it due the complexity of COCO dataset itself, or due to the IMSep while performing T->I. How is the performance while applying original_CLIP+IMSep only on T->I?

[You et. al] Learning visual representation from modality-shared contrastive language-image pre-training. In ECCV, pp. 69–87, 2022.

**Questions:**

Please see weakness.

---

> ### Author Response · Authors · 2024-11-25
> **Response to Reviewer ZbcG**
>
> Thank a lot for your positive comments and recognizing the strengths of our work. Hereby, we wish to provide more clarification to the raised concerns.
>
> **W1.** The extension of the sharing parameters is necessary for increasing the inductive bias towards shared modality encoders, and therefore, reduce the gap. Details about this justification is **provided in L146-160**. Furthermore, in order to further show the effectiveness of the Intra-Modality Separation (IMSep) loss, regardless of sharing parameters, we have conducted an ablation study for this rebuttal and report the results in the main rebuttal. Our results consistently show the effectiveness of the IMSep loss even without sharing parameters. This observation **supports the effectiveness of using the IMSep loss for dual-stream encoders, e.g., using CLIP-ResNet backends**. We have also **added this ablation to the revised paper, Tables 1-5**.
>
> **W2.** Thanks for the great suggestion. As mentioned in response to W1, we have performed this ablation study and provide the results in the main rebuttal and updated the paper accordingly.
>
> **W3.** Regarding the retrieval performance, our hypothesis is that it is mainly due to the complications that arise from the MSCOCO and Flickr datasets. Previous work [1] shows that these dataset include coarse-grained information. In our paper, in Figure 6A, Figure 9 and Figure 10, we also show how all models retrieve samples that are semantically relevant for the queries, but since the texts are coarse-grained, they might not necessarily be the ground truth. Such characteristics of the dataset can particularly affect the evaluation results of AlignCLIP, due to the semantic regularization in IMSep, as it is dependant on the similarity of the texts. Therefore, if the texts include coarse-grained information, AlignCLIP makes the samples closer to each other based on a higher level of semantics and therefore, can retrieve semantically close samples that are not necessary labeled as the ground truth. We suspect that this phenomenon is more noticeable in the retrieval scenario in comparison to classification datasets, as in the classification tasks, classes are semantically more disjoint.
>
> We thank you again for your feedback on our paper. We have included your suggestion for comparing the original_CLIP+IMSep only versus  (original_CLIP+)SharedCLIP in our revised paper. We hope that we have addressed your other questions and concerns sufficiently well and hope that you would consider raising your scores.
>
> **References:**
>
> [1] Chen, Weijing, Linli Yao, and Qin Jin. "Rethinking benchmarks for cross-modal image-text retrieval." Proceedings of the 46th International ACM SIGIR Conference on Research and Development in Information Retrieval. 2023

---

> > ### Comment · Reviewer_ZbcG · 2024-11-25
> > **Thanks for the feedback**
> >
> > Thanks for the substantial feedback from the authors, and the responses have addressed my concerns.

---

### Official Review · Reviewer_J1QT · 2024-10-29

**Soundness:** 3
**Presentation:** 2
**Contribution:** 3
**Rating:** 6
**Confidence:** 4

**Summary:**

The paper introduces AlignCLIP, a novel training strategy that aims to improve the cross-modal alignment of CLIP-like models by addressing the modality gap between image and text in the shared embedding space. The authors propose to share the learnable parameters between the vision and language encoders to align the two modalities more closely, and to add an Intra-Modality Separation objective function that pushes semantically dissimilar image embeddings apart. Authors train from scratch CLIP, a SharedCLIP (with shared learnable parameters between the vision and language encoders), and AlignCLIP on CC12M. Extensive experiments demonstrate that AlignCLIP improves zero-shot classification, robustness to distribution shifts, and multi-modal retrieval tasks compared to the original CLIP.

**Strengths:**

- AlignCLIP demonstrates significant improvements to the original CLIP, with measurable gains in downstream tasks such as zero-shot classification and robustness to distribution shifts.
- The IMSep objective adds a novel approach to handling intra-modality representations, ensuring that semantically dissimilar image embeddings are spread apart without affecting semantically similar pairs.
- The paper provides both quantitative and qualitative analyses (I really appreciate Fig. 5), providing clear insights into improvements in retrieval tasks and the reduction of the modality gap.

**Weaknesses:**

- AlignCLIP’s performance on fine-tuned retrieval tasks shows marginal improvements over SharedCLIP, with SharedCLIP sometimes even outperforming AlignCLIP (see Tab. 5).
- Ablation studies on the key components, while present, could be expanded. In particular, it would be interesting to explore how much the "Pre-trained Sentence Encode" improves the performance and the benefits of using the ImSep loss component. I am afraid that the smaller improvements in such cases could arise just from using an additional pre-trained encoder (i.e. SBERT all-mpnet-base-v2) at training time.

**Additional Consideration**. I noticed several significant errors within the method section Sec. 3.1 and the method figure (Fig. 2) that impact its reliability. In particular, the figure shows:
- *Conv2d Applied to Image Patches*: I believe this is an error stemming from a misunderstanding of the common CLIP practice of using a Conv2d with a stride equal to the image patch size (that is equivalent to applying a standard linear layer to the flattened patches). So depicting a Conv2d after the patches, as the authors proposed, seems incorrect.
- *Max Pooling Applied to Text Embeddings*: Similarly, I suspect that applying max pooling to the text token patches is another mistake. The authors stated: "Following the original CLIP, we use max-pooling for text embeddings." (line 165). However, I believe this results from a misunderstanding of OpenAI's CLIP implementation, which uses an argmax operation to select the EOS token.
These errors impact the clarity and accuracy of the figure and may mislead readers regarding the model architecture.

**Questions:**

- Q1. How would AlignCLIP perform with different backbone architectures besides ViT-B-16? Is the performance consistent with other vision transformers?
- Q2. Could you explain why AlignCLIP’s retrieval performance only marginally improves over SharedCLIP in certain settings? Are there any specific modifications you plan to address this?
- Q3. Conducting an ablation study of CLIP using the ImSep loss would be insightful. If such experiments haven't been performed yet, evaluating whether this approach might outperform or underperform compared to AlignCLIP in zero-shot classification and cross-modal retrieval could offer meaningful comparisons.
- Q4. While mitigating the modality gap is a promising approach to enhancing cross-modal alignment, it might be worth discussing its effectiveness given that the standard definition of the "modality gap" [1]—the difference between the centroids of the two modalities—can be zero even if individual image-text pairs aren't perfectly aligned. Clarifying why this approach remains effective under these conditions could strengthen the understanding of its impact.

[1] Mind the Gap: Understanding the Modality Gap in Multi-modal Contrastive Representation Learning (https://arxiv.org/abs/2203.02053)

---

> ### Author Response · Authors · 2024-11-25
> **Response to Reviewer J1QT (part 1/2)**
>
> We appreciate your close attention to details and providing insightful reviews.
>
> Regarding your proposed corrections, we checked the source code again and applied the corrections in the revised version of the paper.
>
> In the following, we provide responses to your raised questions.
>
> **Q1.**  To answer this question, we **expanded our experiments using the ViT-S-16 backend** and provide the results in here. In general, we observe the same trend of improvements, as observed with ViT-B-16, which shows that AlignCLIP is effectively reducing the gap while improving downstream tasks for at least **two transformer backends**.
>
> Results of the **alignment score**:
> | **Model**  | **CC3M**   | **MSCOCO** | **ImageNet-1K** | **CIFAR-100** | **CIFAR-10** |
> |------------|------------|------------|-----------------|---------------|--------------|
> | CLIP   	| 0.4  	|0.44   	|0.38        	|  0.58    	|0.36      	|
> | SharedCLIP |  0.58 	|0.61   	|  0.57      	| 0.56     	| 0.57    	|
> | AlignCLIP  |0.67 | 0.69  |  0.66  	| 0.64 	|0.65	|
>
> Results of **zero-shot image classification**:
> | **Model**  | **ImageNet1K-top1** | **ImageNet1K-top5** | **CIFAR-100-top1** | **CIFAR-100-top5** | **CIFAR-10-top1** | **CIFAR-10-top5** | **Flowers-102-top1** | **Flowers-102-top5** | **StanfordCars-top1** | **StanfordCars-top5** |
> |------------|---------------------|---------------------|--------------------|--------------------|-------------------|-------------------|----------------------|----------------------|-----------------------|-----------------------|
> | CLIP   	|  30.5          	|   57.7         	|  25.3         	|  55.1         	|   57.5       	|  94.9        	|  16.8             	| 37.6            	| 9.2             	|  32            	|
> | SharedCLIP | 29.8           	| 56.5           	| 27.6          	| 56.7          	|61.3          	|95.8          	| 17.7            	|37             	|8.8               	| 30.9             	|
> | AlignCLIP  | 30.8       	|57.8        	| 27.8       	| 57.5      	|63.2      	|95.9      	| 20.6             	| 38.8             	| 9.5         	|32.9          	|
>
> Results of **image classification with distribution shifts**:
> | **Model**  | **ImageNetV2-top1** | **ImageNetV2-top5** | **ImageNet-R-top1** | **ImageNet-R-top5** | **ImageNet-A-top1** | **ImageNet-A-top5** | **ImageNetSketch-top1** | **ImageNetSketch-top5** |
> |------------|---------------------|---------------------|---------------------|---------------------|---------------------|---------------------|-------------------------|-------------------------|
> | CLIP   	| 26.1           	| 52           	| 36.3           	| 63           	|5.7             	|22.4            	|18                 	|39.6                	|
> | SharedCLIP | 25.1           	|50.3            	|36.5            	|62.8            	|5.4             	|22.2            	| 18               	| 39.6           	|
> | AlignCLIP  |26.4        	|52.4        	|38.1        	|64.5       	|6.3           	| 23.4       	| 18.6           	| 39.6           	|
>
> We have also provided the results of **retrieval experiments** in the revised version of the paper  **Section A.6, Tables 10-13**.

---

> > ### Author Response · Authors · 2024-11-25
> > **Response to Reviewer J1QT (part 2/2)**
> >
> > **Q2.** Regarding AlignCLIP’s performance in retrieval tasks, we suspect that it is mainly due to the complications that arise from the MSCOCO and Flickr datasets as they include coarse-grained information. This issue has been studied in the previous work [1] and in our paper, in Figure 6A, Figure 9 and Figure 10, we also show how all models retrieve samples that are semantically relevant for the queries, but might not necessarily be the ground truth, as the texts include coarse-grained information. Such characteristics of the dataset can particularly affect the evaluation results of AlignCLIP, due to the semantic regularization in IMSep, as it is dependant on the similarity of the texts. Therefore, if the texts include coarse-grained information, AlignCLIP makes the samples closer to each other based on a higher level of semantics and therefore, can retrieve semantically close samples that are not necessary labeled as the ground truth. We suspect that this phenomenon is more noticeable in the retrieval scenario in comparison to classification datasets, as in the classification tasks, classes are semantically more disjoint. In future work, we plan to test AlignCLIP on the MSCOCO-FG and Flickr30k-FG datasets proposed in [1], which include more fine-grained samples and test our hypothesis.
> >
> > **Q3.** Regarding the ablation studies, we have **expanded our ablation to include the results of adding IMSep loss** without sharing the transformer architecture and have added the results in the **main rebuttal**. We have also **added this ablation to the revised paper, Tables 1-5**. Furthermore, if you have suggestions for more ablation studies, we would be happy to provide them.
> >
> > **Q4.** We would sincerely appreciate it if you could clarify this question with more details.
> >
> > **References:**
> >
> > [1] Chen, Weijing, Linli Yao, and Qin Jin. "Rethinking benchmarks for cross-modal image-text retrieval." Proceedings of the 46th International ACM SIGIR Conference on Research and Development in Information Retrieval. 2023

---

> ### Comment · Reviewer_J1QT · 2024-11-26
>
> I thank the authors for addressing Q1, Q2, and Q3, and I apologize for any confusion regarding Q4.
>
> To clarify **Q4**: *Mitigating* the *modality gap* (i.e., reducing the centroid difference between the two modalities) is discussed in the manuscript as a means to improve performance. However, I emphasize that achieving a zero gap does not necessarily guarantee perfect pair-wise alignment. Could you elaborate on your motivation for focusing on mitigating the gap, or explain when and why you believe this could be beneficial or not?
>
> As requested for additional suggestions, please see **W2** and **Additional Considerations**, which remain unanswered.

---

> > ### Author Response · Authors · 2024-11-26
> > **Response to the Comment by Reviewer J1QT**
> >
> > Thanks for providing more clarifications.
> >
> > **Q4.** Please note that in contrast to previous work [2, 3], that define the gap only by the distance of the centroids of each modality, **our definition** of the **alignment score in Eq. 5 of the paper** is indeed **measuring the average pairwise image–text alignment**, i.e., the **average of the cosine distances of each paired image–text**. Therefore, the alignment scores presented in **Table 1** of the submitted paper measure the **pairwise alignment** of the paired image–text samples and we show that AlignCLIP achieves a noticeable alignment, and therefore, reduction in the gap.
> >
> > > Could you elaborate on your motivation for focusing on mitigating the gap, or explain when and why you believe this could be beneficial or not?
> >
> > Our motivation for mitigating the gap is to enable the creation of a meaningful and **semantically well-structured** cross-modal embedding space in which **semantically similar samples are fairly closely aligned** with each other, **regardless of their modalities**, as they **represent the same semantics**. E.g., a visual image of a red apple and its pairing caption “a red apple” convey the same semantics, therefore, aligning their embeddings in the embedding space leads to the **creation of meaningful multi-modal semantic clusters**, and therefore, **can improve the downstream tasks**. In our paper, using the definition of the alignment score in **Eq. 5, we measure the distance of the paired image–texts, carrying the same semantics**.
> >
> > **W2.** Sorry for the misunderstanding. Our understanding was that your mention of the ablation study in W2 is related to the ablation study requested in Q3, to which we provided the response.
> >
> > > In particular, it would be interesting to explore how much the "Pre-trained Sentence Encode" improves the performance and the benefits of using the ImSep loss component.
> >
> > To this end, we have already **provided this ablation study in the submitted paper in Table 6\.** This ablation shows that **using the pre-trained sentence encoder** for regularizing the IMSep loss, i.e., **re-scaling** the image–image similarities in Eq. 10, is indeed **effective**.
> >
> > > I am afraid that the smaller improvements in such cases could arise just from using an additional pre-trained encoder (i.e. SBERT all-mpnet-base-v2) at training time.
> >
> > As Table 6 shows, the effectiveness of the IMSep loss is dependent on the regularization and the re-scaling mechanism defined in Eq.9-10. The re-scaling, itself, requires a source of “knowledge” that already “knows” which samples are similar or dissimilar. In our work, we have chosen the pre-trained sentence encoder to be this source of knowledge. Therefore, it is indeed recommended to choose a pre-trained encoder that does include the sufficient knowledge. For our experiments, we chose the SBERT `all-mpnet-base-v2` checkpoint, since this checkpoint achieved the **top rank** results in the leaderboard for the **task of semantic text similarly**, which is the core of Eq. 9 in our paper. The semantic text similarity leaderboard can be found at the official documentation of Sentence Transformers available at \[1\].
> >
> > We hope that our response has addressed your concern sufficiently well. If otherwise, could you please clarify more concretely which other ablation studies you wish us to perform?
> >
> > **Additional Considerations**
> >
> > Apologies for not being clear in our first response. We did already check the source code from the Open CLIP implementation and applied the corrections in **Figure 2 of the revised paper**.
> >
> > We thank you again for your constructive feedback and questions. We hope that we have addressed your concerns sufficiently well and hope that you consider raising your scores.
> >
> > **References:**
> >
> > [1] <https://sbert.net/docs/sentence_transformer/pretrained_models.html>
> >
> > [2] Liang, Victor Weixin, et al. "Mind the gap: Understanding the modality gap in multi-modal contrastive representation learning." Advances in Neural Information Processing Systems 35 (2022): 17612-17625.
> >
> > [3] Schrodi, Simon, et al. "Two Effects, One Trigger: On the Modality Gap, Object Bias, and Information Imbalance in Contrastive Vision-Language Representation Learning." arXiv preprint arXiv:2404.07983 (2024).

---

> > > ### Comment · Reviewer_J1QT · 2024-11-27
> > >
> > > **Q4** has been thoroughly addressed.
> > >
> > > **Figure 2** seems fixed, but I think there is still an inconsistency at **line 165**: "Following the original CLIP, we use max-pooling for text embeddings.". I suggest authors to double-check it.
> > >
> > > My main concern remains the fairness of the comparison. Using such a text encoder during training might help AlignCLIP train faster, which could explain its slight performance improvement over CLIP given the limited training resources. Without this auxiliary sentence encoder, the performance benefit might be negligible.
> > >
> > > Overall, given the clarifications of the authors, *I will increase my rating from 5 to 6*.

---

> ### Author Response · Authors · 2024-11-27
> **Response to the Comment by Reviewer J1QT**
>
> Thanks for the discussions. We have now updated **Line 165** in the revised manuscript as follows:
>
> > Following the original CLIP, we use the [EOS] token pooling for text embeddings and the [CLS] token embedding for the image embedding.

---

### Official Review · Reviewer_XfLj · 2024-10-30

**Soundness:** 3
**Presentation:** 4
**Contribution:** 3
**Rating:** 8
**Confidence:** 3

**Summary:**

This paper proposes AlignCLIP, a solution to the text-image alignment issue in CLIP. Specifically, AlignCLIP (1) shares encoder parameters and (2) enhances cross-modal alignment without disrupting embedding distances by introducing IMSep. IMSep particularly adjusts the unimodal embeddings, placing semantically similar image embeddings close but dissimilar image embeddings apart. Experimental results demonstrate substantial improvements in zero-shot classification, robustness to distribution shifts, and multi-modal retrieval, highlighting AlignCLIP's superior alignment and performance over existing CLIP models.

**Strengths:**

- AlignCLIP effectively addresses the text-image alignment issue in CLIP, offering meaningful performance gains over existing methods.

- I like the idea of IMSep, adjusting unimodal embeddings for enhancing cross-modal alignment.

- The method has been verified under various tasks-- especially achieving gains both in zero-shot classification and retrieval seem meaningful.

**Weaknesses:**

- I am not very confident in the evaluation settings. Are all competing models trained on the same dataset? If not, how is performance comparability ensured?

- The ideas are interesting; however, this still falls within contrastive learning. Could you strengthen the value of your method by connecting it to existing work? While I believe your approach is new, a clearer link to prior work would enhance its contribution.

**Questions:**

Please see the above weakness.

---

> ### Author Response · Authors · 2024-11-25
> **Response to Reviewer XfLj**
>
> Thanks a lot for your great feedback on our paper and acknowledging the strength of our method and extensive experimental results.
>
> > Are all competing models trained on the same dataset?
>
> **Yes**, for all comparisons, the same dataset has been used. I.e., in Tables 1-6, all models have been trained using the CC12M dataset. In Table 7, all models have been trained using the CC3M dataset.
>
> > While I believe your approach is new, a clearer link to prior work would enhance its contribution.
>
> To the best of our knowledge, our proposed semantically-regualized InfoNCE contrastive loss for intra-modality separation (IMSep)  is quite novel and has not been proposed in prior work.

---

> > ### Comment · Reviewer_XfLj · 2024-11-27
> >
> > Authors,
> > Thanks for your effort. Considering all feedback made by other reviewers and my judgement, I keep my rating as 8 (Accept).
> >
> > BTW, I am surprised that only I give it an "Accept" score. The technical value of this paper, in terms of innovative ideas and presentation quality, is pretty high, the best one in my batch.

---

### Official Review · Reviewer_6cpP · 2024-10-31

**Soundness:** 2
**Presentation:** 3
**Contribution:** 2
**Rating:** 6
**Confidence:** 4

**Summary:**

This paper introduces AlignCLIP, a variant of the CLIP model aimed at reducing the modality gap between image and text embeddings in cross-modal learning. The authors propose two main modifications: SharedCLIP, which shares the learnable parameters of vision and language encoders, and Intra-Modality Separation (IMSep), an additional objective that regularizes the distances within each modality based on semantic dissimilarity. Through extensive experiments, the authors demonstrate improvements in cross-modal alignment and zero-shot performance, with AlignCLIP consistently outperforming baseline CLIP on a range of benchmarks.

**Strengths:**

- The authors address a meaningful challenge in multimodal learning, i.e., the modality gap, which affects alignment quality in embedding spaces.
- The paper evaluates performance on various benchmarks (e.g., zero-shot classification and retrieval tasks) and robustness against distributional shifts, making the findings broad in scope.
- Visualizations, like the DOSNES projection and alignment score plots, effectively show the improvements in modality alignment and embedding spread across models.

**Weaknesses:**

- Both SharedCLIP and IMSep primarily extend existing contrastive learning techniques. Sharing parameters between encoders is a known approach, and the IMSep loss largely repurposes InfoNCE without substantial modifications. The paper’s novelty, therefore, is limited.
- Empirical comparisons with relevant baselines are lacking, as AlignCLIP is only compared with the original CLIP (and SharedCLIP, also introduced in this paper), omitting several cited "naive" approaches for modality gap reduction.
- For example, the paper repeatedly claims that naive alignment approaches "likely" negatively impact downstream zero-shot performance and backs up this with a theoretical argument, but no qualitative or quantitative results are provided to show that this is indeed the case.
- Table 7 also compares AlignCLIP with two other CLIP variations, but from what I understand, these variations were not designed to mitigate the modality gap.
- In L198-200, the text claims the denominator gets strongly minimized, but the next sentence in L200-202 says it is not strongly minimized. This seems to be a mistake.
- L269 says "In AlignCLIP, we set $\alpha=1$ and $\beta=0.5$", $\beta$ is never introduced in the paper. Moreover, in appendix A.2, it says $\alpha=0.5$ was used, which contradicts this line.

**Questions:**

- The paper states that naive shifts harm downstream performance due to distorted relative distances. Could the authors provide empirical evidence or experiments to support this claim? e.g., Liang et al. (2022)
- The choice of hyperparameters, such as the weight of the IMSep objective in total loss($\alpha$), appears arbitrary. Did the authors conduct sensitivity analyses on these hyperparameters, and if so, could they report their findings?
- While alignment scores improve, does this necessarily translate to enhanced downstream performance? Could the authors discuss specific cases where improved alignment directly correlates with task gains?
- How significant are the alignment improvements in practical, real-world applications? For instance, does better alignment consistently yield qualitatively superior results in retrieval scenarios?
- Do you expect your method to scale easily to larger backbones (e.g., ViT-L or ViT-H) using a dataset like LAION-400M instead of CC12M?
- How is the pre-trained sentence encoder (SBERT) used to calculate semantic distance? Is it robust across various domains and vocabularies in the dataset?

---

> ### Author Response · Authors · 2024-11-25
> **Response to Reviewer 6cpP (part 1/3)**
>
> Thanks for taking the time to review our paper. In the following, we provide more clarifications to the raised points.
>
> **W1.** We would like to emphasize that our proposed **semantically-regularized IMSep loss is a novel approach in contrastive learning**, allowing for controlling the harshness of the optimization of the denominator in the InfoNCE loss. To the best of knowledge, this objective function has not been proposed before and can be utilized in other scenarios where such regularization in the InfoNCE loss is required.
>
> **W2-W4.** Thanks for your input. We implemented the naive approaches introduced in [1, 2] and updated our SOTA comparison, respectively. The updated results are as follows:
>
> | Model  	| ImageNet1K-top1 | ImageNet1K-top5 | CIFAR-100-top1 | CIFAR-100-top5 | CIFAR-10-top1 | CIFAR-10-top5 | ImageNet-A-top1 | ImageNet-A-top5 | Alignment |
> |------------|-----------------|-----------------|----------------|----------------|---------------|---------------|-----------------|-----------------|-----------|
> | CLIP   	| 16.5        	| 34.1        	| 23.7       	| 49.7       	| 51.3      	| 92.7      	| 4           	| 15.8        	| 0.36  	|
> | CyCLIP 	| 15          	| 32          	| 21.8       	| 48         	| 44.7      	| 85.4      	| 3.3         	| 13.6        	| 0.32  	|
> | Naive Shifting 	| 16.6          	| 34.6          	| 24.5       	| 52.5         	| 51.2      	| 91.2      	| 43.3         	| 68.8        	| 0.16 (with lambda=0.5) |
> | SharedCLIP | 17.4        	| 34.9        	| 26         	| 52.1       	| 57        	| 92.9      	| 3.7         	| 14.5        	| 0.45  	|
> | AlignCLIP  | 17.6        	| 34.9        	| 26.3       	| 54         	| 57.9      	| 93.7      	| 4.2         	| 16.1        	| 0.63  	|
>
> We observe that AlignCLIP achieves the best results in almost all the tasks. However, in classification with distribution shift for ImageNet-A, we notice that naive shifting approach remarkably improves the classification accuracy. We attempted at getting some insights from the original papers \[1, 2\] and notice that the distribution shift experiments have not been studied in neither \[1, 2\]. **We have included this comparison in the revised version of our paper as well**.
>
> **W5.** We respectfully ague that L198-202 are indeed correct. In L198-200, we describe the situation where the samples are semantically close (L199), and in L200-202, we describe the opposite scenario, where the samples are semantically dissimilar (L201).
>
> **W6.** Thanks for pointing out the typo in L269. We have corrected this in the revised version of the paper.

---

> > ### Author Response · Authors · 2024-11-25
> > **Response to Reviewer 6cpP (part 2/3)**
> >
> > In response to your questions:
> >
> > **Q1.** We respectfully draw your attention to Section A.1 in our paper, **where we have provided theoretical argument** on how the shifting approach utilized in \[1, 2\] has changed the relative cross-modal distances, and therefore, resulted in incorrect retrieval scenario after the shift. Furthemore, please refer to our response to W1-W4 for a **quantitative comparison**. We have also **included this comparison in the revised version of the paper**.
> >
> > **Q2.** We chose the value of alpha based on **hyperparameter sweeping** and observed that alpha=0.5 achieves the best downstream performances, especially for multi-modal retrieval experiments.
> >
> > **Q3.** **Yes**, our extensive experiments provided in the paper shows that AlignCLIP reduces the modality gap and improves various zero-shot and fine-tuned downstream tasks.
> >
> > **Q4 and Q5.** In order to answer these, we tested our models using the ViT-S-16 backend and **observed the same trend of improvements** as in ViT-B-16. In the following, we provide the corresponding results.
> >
> > Results of the **alignment score**:
> > | **Model**  | **CC3M**   | **MSCOCO** | **ImageNet-1K** | **CIFAR-100** | **CIFAR-10** |
> > |------------|------------|------------|-----------------|---------------|--------------|
> > | CLIP   	| 0.4  	|0.44   	|0.38        	|  0.58    	|0.36      	|
> > | SharedCLIP |  0.58 	|0.61   	|  0.57      	| 0.56     	| 0.57    	|
> > | AlignCLIP  |**0.67** | **0.69**  | **0.66**  	| **0.64** 	|**0.65**	|
> >
> > Results of **zero-shot image classification**:
> > | **Model**  | **ImageNet1K-top1** | **ImageNet1K-top5** | **CIFAR-100-top1** | **CIFAR-100-top5** | **CIFAR-10-top1** | **CIFAR-10-top5** | **Flowers-102-top1** | **Flowers-102-top5** | **StanfordCars-top1** | **StanfordCars-top5** |
> > |------------|---------------------|---------------------|--------------------|--------------------|-------------------|-------------------|----------------------|----------------------|-----------------------|-----------------------|
> > | CLIP   	|  30.5          	|   57.7         	|  25.3         	|  55.1         	|   57.5       	|  94.9        	|  16.8             	| 37.6            	| 9.2             	|  32            	|
> > | SharedCLIP | 29.8           	| 56.5           	| 27.6          	| 56.7          	|61.3          	|95.8          	| 17.7            	|37             	|8.8               	| 30.9             	|
> > | AlignCLIP  | 30.8       	|57.8        	| 27.8       	| 57.5      	|63.2      	|95.9      	| 20.6             	| 38.8             	| 9.5         	|32.9          	|
> >
> > Results of **image classification with distribution shifts**:
> > | **Model**  | **ImageNetV2-top1** | **ImageNetV2-top5** | **ImageNet-R-top1** | **ImageNet-R-top5** | **ImageNet-A-top1** | **ImageNet-A-top5** | **ImageNetSketch-top1** | **ImageNetSketch-top5** |
> > |------------|---------------------|---------------------|---------------------|---------------------|---------------------|---------------------|-------------------------|-------------------------|
> > | CLIP   	| 26.1           	| 52           	| 36.3           	| 63           	|5.7             	|22.4            	|18                 	|39.6                	|
> > | SharedCLIP | 25.1           	|50.3            	|36.5            	|62.8            	|5.4             	|22.2            	| 18               	| 39.6           	|
> > | AlignCLIP  |26.4        	|52.4        	|38.1        	|64.5       	|6.3           	| 23.4       	| 18.6           	| 39.6           	|

---

> > > ### Author Response · Authors · 2024-11-25
> > > **Response to Reviewer 6cpP (part 3/3)**
> > >
> > > Results of **zero-shot** **multi-modal retrieval** experiments are provided as well:
> > > | **Model**  | **MSCOCO-I2T-R@1** | **MSCOCO-I2T-R@5** | **MSCOCO-I2T-R@10** | **MSCOCO-T2I-R@1** | **MSCOCO-T2I-R@5** | **MSCOCO-T2I-R@10** | **FLICKR-I2T-R@1** | **FLICKR-I2T-R@5** | **FLICKR-I2T-R@10** | **FLICKR-T2I-R@1** | **FLICKR-T2I-R@5** | **FLICKR-T2I-R@10** |
> > > |------------|--------------------|--------------------|---------------------|--------------------|--------------------|---------------------|--------------------|--------------------|---------------------|--------------------|--------------------|---------------------|
> > > | CLIP   	| 30.4          	| 56.9            	| 67.4           	| 19.9          	|42           	|53.7            	|52.1           	| 77.9           	|85.8            	| 38.2          	| 66            	| 76.1          	|
> > > | SharedCLIP | 31.3          	| 57          	| 68.1            	|  20.3         	| 42.2      	|54.1        	|52.1        	|  80.9     	|   87.4     	|38.6           	|   66.4          	|76.1       	|
> > > | AlignCLIP  |31.8    	| 57.1      	|68.1        	|19.9       	|42.2       	| 53.9           	| 52.7          	|80.6          	|87.2        	|38.7       	| 66.6       	| 76.4       	|
> > >
> > > Lastly, the results of **fine-tuned** **multi-modal retrieval** experiments are provided in the following table:
> > > | **Model**  | **MSCOCO-I2T-R@1** | **MSCOCO-I2T-R@5** | **MSCOCO-I2T-R@10** | **MSCOCO-T2I-R@1** | **MSCOCO-T2I-R@5** | **MSCOCO-T2I-R@10** | **FLICKR-I2T-R@1** | **FLICKR-I2T-R@5** | **FLICKR-I2T-R@10** | **FLICKR-T2I-R@1** | **FLICKR-T2I-R@5** | **FLICKR-T2I-R@10** |
> > > |------------|--------------------|--------------------|---------------------|--------------------|--------------------|---------------------|--------------------|--------------------|---------------------|--------------------|--------------------|---------------------|
> > > | CLIP   	|38.2           	|66           	|76.3             	| 25          	| 50.5          	|62.8            	|61           	|84.8           	| 91.1           	|45.2           	| 72.1          	|82              	|
> > > | SharedCLIP |38.5           	|66           	|76.8            	| 25.2          	| 51.1        	|  63.4      	|61.1       	|  85.1     	| 91.7           	|45.4           	| 73.4          	|  82.2          	|
> > > | AlignCLIP  | 39      	|  66.8     	|76.4        	| 25.4      	| 50.8          	|  63          	| 61.1            	| 85.4      	|  91.6      	|45.5       	| 72.8      	|  82      	|
> > >
> > > We have also **included the result of this new experiment in the revised version of the paper, Section A.6, Tables 10-13**.
> > >
> > > **Q6.** In L207-2011 and Eq. 9, we have explained in detail how the pre-trained sentence encoder calculates the similarities. Since this pre-trained encoder is acting as a “knowledge” supervision, it is indeed recommended to chose a pre-trained encoder that does include the domain-specific knowledge in respective domain.
> > >
> > > Thanks for taking the time to review our paper and providing your feedback. We hope that we have addressed your concerns and you would consider increasing your scores.
> > >
> > > **References:**
> > >
> > > [1] Liang, Victor Weixin, et al. "Mind the gap: Understanding the modality gap in multi-modal contrastive representation learning." Advances in Neural Information Processing Systems 35 (2022): 17612-17625.
> > >
> > > [2] Schrodi, Simon, et al. "Two Effects, One Trigger: On the Modality Gap, Object Bias, and Information Imbalance in Contrastive Vision-Language Representation Learning." arXiv preprint arXiv:2404.07983 (2024).

---

> > > > ### Comment · Reviewer_6cpP · 2024-11-27
> > > >
> > > > Thanks for the detailed feedback. After reading the discussion with other reviewers and examining additional results provided by the authors, I tend to view this paper more positively. I will raise my score accordingly.

---

### Official Review · Reviewer_mh82 · 2024-11-01

**Soundness:** 2
**Presentation:** 3
**Contribution:** 2
**Rating:** 6
**Confidence:** 4

**Summary:**

This paper proposes AlignCLIP to reduce the modality gap in CLIP embedding space. The method leverages a shared transformer to align the image and text embeddings. An additional loss function is also introduced to separate semantically distinct unimodal embeddings. Experimental results show that the proposed method can reduce the modality gap and improve the performance on downstream tasks.

**Strengths:**

1. Investigating the multimodal embedding space of vision-language models is an interesting topic.
2. The proposed framework seems easy to implement while still being very effective.

**Weaknesses:**

1. The technical contribution appears limited, as prior work has also leveraged the shared transformer framework [1, 2].

2. The stated goal of this work is to reduce the modality gap within the CLIP embedding space. So basically, the approach should rely on the pre-trained CLIP embedding space. However, the proposed method utilizes a single-encoder framework, which is completely different from CLIP’s two-tower architecture. This raises questions about whether the obtained embedding space is relevant to the CLIP embedding space, which could benefit from further clarification.

3. To my knowledge, the effect of the modality gap on downstream task performance remains an open question. In some cases, a larger modality gap can actually improve performance on certain datasets [3]. This paper could be strengthened by including a more in-depth discussion or insights.




[1] You et al. Learning Visual Representation from Modality-Shared Contrastive Language-Image Pre-training. \
[2] Girdhar et al. OMNIVORE: A Single Model for Many Visual Modalities. \
[3] Liang et al. Mind the Gap: Understanding the Modality Gap in Multi-modal Contrastive Representation Learning.

**Questions:**

please see the weaknesses.

---

> ### Author Response · Authors · 2024-11-25
> **Response to Reviewer mh82**
>
> Thanks for sharing your feedback and raising your concerns. In the following, we provide individual responses to the weaknesses you have mentioned.
>
> W1. Regarding the technical novelty, please note that AlignCLIP’s **contribution is twofold**: 1\. **Sharing the transformers**, which we have acknowledged in the paper that has been proposed by previous work, and we only extend it to include all parameters that can be shared, in order to increase the inductive bias towards shared cross-modal encoders, 2\. **A novel semantic-regularized intra-modality separation** InfoNCE loss function proposed in Section 3.2, i.e., IMSep. In our rebuttal, we have provided the ablation of using IMSep loss without sharing parameters, in order to show the effectiveness of IMSep, regardless of the sharing architecture. The results are provided in the main rebuttal and in the revised version of the paper, Tables 1-5. This ablation study shows that **our novel IMSep objective function is effective in improving all of the downstream performances while reducing the modality gap regardless of sharing the transformers**.
>
> W2. In order to test this, we performed the ablation of using IMSep on the dual encoder scenario, i.e., without sharing the transformer encoder, and reported the results in the main rebuttal and revised Tables 1-5, accordingly. We observed the **improvement of the gap while enhancing all downstream performances when using the two-tower architecture**.
>
> W3. Thanks for the suggestion. **We have already provided this analysis in our submitted paper**. As mentioned in **Section 1** of our paper, the work you mentioned (\[3\]) has used naive shifting for closing the gap which can damage the cross-modal distances. In **Section A.1** of our submitted paper, we have **provided a theoretical argument** of how this shifting can change the relative cross-modal distances. As a result, authors in \[3\] could not achieve a consistent outcome from reducing the gap. However, in AlignCLIP, we reduce the gap by enforcing an inductive bias towards shared modality encoders as well as training a semantic separation of the unimodal embeddings, which results in a more meaningful structure of the embedding space, as it considers the semantic relationships.
>
>
> We thank you again for your reviewing our paper and providing feedback. We hope that we have addressed them sufficiently well. We would be happy to provide more clarifications and appreciate it if you consider raising your scores.

---

> > ### Comment · Reviewer_mh82 · 2024-11-26
> >
> > Thanks for the authors' responses. I hope the content in Section A.1 can be organized into the main body of the final version. It will help the reader better understand the motivation of this work.
> >
> > Overall, the responses have addressed my concerns. I will increase my rating.

---

### Official Review · Reviewer_WiMx · 2024-11-02

**Soundness:** 3
**Presentation:** 3
**Contribution:** 3
**Rating:** 6
**Confidence:** 4

**Summary:**

The paper proposes a method to mitigate the modality gap in the representational space of CLIP. The authors introduce two modifications to CLIP (termed AlignCLIP) to achieve this: (1) Sharing the weights of the vision and text transformers in CLIP (termed ‘SharedCLIP’), and (2) using a new Intra-modality separation loss, which encourages separation between images in CLIP space while respecting some of the semantics of the similarities between the images.

The paper’s experiments study the effects of the modifications on a range of downstream tasks, and demonstrate that the AlignCLIP reduces the modality gap and improves zero-shot image classification, linear probe accuracy, robustness to distribution shifts, and multi-modal retrieval accuracy.

**Strengths:**

**Well motivated paper:** Understanding and mitigating the modality gap is an important direction in CLIP and its variants, which are ubiquitously used in many applications

**Clearly written:** Clear flow of ideas, motivations well explained

**Comprehensive ablation studies:** The authors suggested two directions (SharedCLIP + IMSep loss) for attempting to reduce the modality gap, and the effects of both SharedCLIP and SharedCLIP + IMSep loss were tested. Further, the impact of enforcing intra-modality separation on images, texts and their combination was tested individually. Finally, the effect of the rescaling mechanism to control separation of similar images in the batch was also tested individually.

**Weaknesses:**

**Potentially limited applicability:** The approach of AlignCLIP is only applicable where transformer architecture can be used for both modality encoders. This may limit the applicability to other multimodal models with a modality gap using non-transformer architectures: The authors could measure the performance of the proposed IMSep loss without the SharedCLIP architecture, as the IMSep loss could be applied to any non-transformer architecture. (See Question 2)

**Unfair comparison to baselines:** I appreciate the authors’ efforts to have a comparison with popular baselines in Table 7. However, as far as I understand, the numbers for CLIP, CyCLIP, and DFSep in Table 7 were taken from the DFSep paper [a] . However, important hyperparameters that are used to train CLIP, CyCLIP, and DFSep in [a] are different to what the authors used to train SharedCLIP and AlignCLIP in their paper. As a result, these comparisons may not be fair. For instance, authors in [a] trained their models for 64 epochs using batch size of 128, and used 1024 dimensional features. Whereas authors of this paper trained for 30 epochs using 512 batch size, and 768 dimensional features. Further, CLIP models in [a] used ResNet-50 as the image encoder, whereas the authors of this paper used ViT-B-16 backend.

- I would recommend that the authors either re-implement CyCLIP and DFSep using the same hyperparameters as in their experiments, or repeat their experiments using the same hyperparameters as used in CyCLIP and DFSep in [a].
- In either case, these differences should be clearly stated in the paper.

[a] Q. Jiang et al., “Understanding and Constructing Latent Modality Structures in Multi-Modal Representation Learning,” in 2023 IEEE/CVF Conference on Computer Vision and Pattern Recognition (CVPR), Vancouver, BC, Canada: IEEE, Jun. 2023, pp. 7661–7671. doi: 10.1109/CVPR52729.2023.00740.

**Questions:**

1. **Why did the authors use “photo of {caption}” prompt in multi-modal retrieval, instead of just the caption itself?**
Were there any intuitions the authors had about this approach?

2. **Did the authors check performance using just the IMSep loss, without using the architecture of SharedCLIP?**
This would allow the approach to be used regardless of encoder architecture.

3. **Did the authors track how the learnable temperature parameter changes during training in AlignCLIP?** Low temperature is known to be an important factor in the emergence of the modality gap.  Therefore, it would be interesting to note the trend and/or the final value of learned temperature in AlignCLIP.

---

> ### Author Response · Authors · 2024-11-25
> **Response to Reviewer WiMx**
>
> Thank you for the insightful comments and close attention to details. We very much appreciate the raised points.
>
> > W.1: Potentially limited applicability
>
> Thank you for the comment. As you suggested, **we performed a new ablation study for the case of using IMSep loss without sharing parameters**, which **reduces the dependency of our method on the shared transformer architecture**. We have provided the results in the main rebuttal and show that **using the IMSep loss noticeably improves the alignment as well as the downstream performances**.
>
> > W2: Comparison to baselines
>
> Thanks for your valuable input. We implemented your first suggestion and trained CLIP and CyCLIP models ourselves, with the same hyperparameters as our model’s and report the results:
> | Model  	| ImageNet1K-top1 | ImageNet1K-top5 | CIFAR-100-top1 | CIFAR-100-top5 | CIFAR-10-top1 | CIFAR-10-top5 | ImageNet-A-top1 | ImageNet-A-top5 | Alignment |
> |------------|-----------------|-----------------|----------------|----------------|---------------|---------------|-----------------|-----------------|-----------|
> | CLIP   	| 16.5        	| 34.1        	| 23.7       	| 49.7       	| 51.3      	| 92.7      	| 4           	| 15.8        	| 0.36  	|
> | CyCLIP 	| 15          	| 32          	| 21.8       	| 48         	| 44.7      	| 85.4      	| 3.3         	| 13.6        	| 0.32  	|
> | SharedCLIP | 17.4        	| 34.9        	| 26         	| 52.1       	| 57        	| 92.9      	| 3.7         	| 14.5        	| 0.45  	|
> | AlignCLIP  | **17.6**        	| **34.9**        	|**26.3**       	| **54**         	| **57.9**      	| **93.7**      	| **4.2**         	| **16.1**        	| **0.63**  	|
>
> Our new comparisons show that the **AlignCLIP model achieves even stronger results** **in comparison to SOTA models**.
>
> As for the DFSep model, since the authors have not published their source code, given the limited time frame, we had to carefully prioritize our tasks. Nevertheless, we employed your second suggestion for this model, and have added more description in the caption of Table 7 to be more fair.
>
> ## In response to your questions:
>
> Q1. The usage of this prompt is encouraged by the **original CLIP** paper as stated in the supplementary material of \[1\], **Section H.1**:
>
> *For both these datasets we prepend the prompt “a photo of” to the description of each image which we found boosts CLIP’s zero-shot R@1 performance between 1 and 2 points.*
>
> Therefore, we applied the same prompt template and also observed a noticeable boost in the zero-shot performances.
>
> Q2. Thanks for your valuable suggestion. We **expanded our experiments to add this ablation study** and reported the results in the main rebuttal. We have also **included these result in the revised version of the paper, Tables 1-5**.
>
> Q3. Thanks for suggesting this valuable analysis. **Yes**, we have tracked the temperature values during the training . As mentioned in Section A.2 in the paper, the initial temperature value was set to 0.07 for all models. The **final learned temperature** value for the **original CLIP is 0.015**, for **sharedCLIP is 0.011**, and for **AlignCLIP is 0.010**. This concludes that AlignCLIP, while having the least temperature value at the last epoch, achieves the least modality gap, showing the effectiveness of our proposed refinements.
>
> We have tracked the temperature value at each step by logging the logit scale which is the inverse of the temperature value, i.e., 1/temperature. Furthermore, we have **provided the plots for each model in Figure 11 in Section A.5** in the revised version of the paper.
>
> We thank you again for your insightful comments and questions. **We have included all of them in our revised paper** and wish that we have addressed them sufficiently well. We would be happy to provide more clarifications and appreciate it if you consider raising your scores.
>
> **References:**
> [1] Radford et al., Learning Transferable Visual Models From Natural Language Supervision (<http://proceedings.mlr.press/v139/radford21a>)

---

> > ### Comment · Reviewer_WiMx · 2024-11-27
> > **Response to authors**
> >
> > I thank the authors for their detailed response.
> >
> > W1: Upon reviewing the additional results, I can see that IMSep loss without sharing parameters seems to improve performance over CLIP baseline.
> >
> > W2: I appreciate the authors re implementing CyCLIP and CLIP as suggested. As for DFSep, I would suggest further clarifying the differences in the training hyperparameters used for DFSep in the paper.
> >
> > All of my questions have been sufficiently answered. I will increase my rating.

---

> ### Author Response · Authors · 2024-11-27
> **Response to the Comment of Reviewer WiMx**
>
> W2. We have now updated the manuscript and added further clarifications in L507-514 of the revised manuscript as follows:
>
> > For a fair comparison with CLIP andCyCLIP, we re-train these models with the same hyperparameters as SharedCLIP and AlignCLIP. Furthermore, we train versions of SharedCLIP and AlignCLIP with the CC3M dataset and compare the resulting alignment scores as well as the downstream performance in Table7. As for the DFSep model, we report the numbers directly from the paper [Jiangetal.(2023)]. This model uses a ResNet50 backend for the vision encoder and the BERT model for encoding text. In contrast to the rest of the models compared in Table7, DFSep is trained with the batch size of 128, embedding dimensionality of 1024, learningrate of 5e-4 and warmup value of 10000.
>
> Thanks for your discussions and for considering to increase the score.

---

> > ### Author Response · Authors · 2024-12-02
> > **Increasing the rating**
> >
> > > All of my questions have been sufficiently answered. I will increase my rating.
> >
> > Thank you so much for acknowledging that your questions have been fully addressed. We truly value your thoughtful review and the time you’ve taken to engage with our work.
> >
> > We noticed that the current score still reflects the initial rating of 6, and we wanted to kindly check if this aligns with your intention to increase it.
> >
> > We appreciate your consideration.

---

### Official Review · Reviewer_3Rs9 · 2024-11-03

**Soundness:** 2
**Presentation:** 4
**Contribution:** 3
**Rating:** 6
**Confidence:** 3

**Summary:**

This paper proposes the modality gap problems existing in CLIP and shows the multi-modalities distribusion characteristics. To handle this problem, They proposed a novel method named AlignCLIP. The proposed AlignCLIP mitigates the multi-modal gaps with sharing the parameters between all modalities. To align the text-image and image-image well, they also propose the intra-modality separation objective function. Then, they have provided extensive experiment evidance to prove the effectiveness of  AlignCLIP.

**Strengths:**

This paper shows the multi-modalities distribusion characteristics and proposes two novel idea to mitigate such modal gap. This is a clear and well-writen paper, easy to follow. The experiments for ablation is sufficient. This idea of AlignCLIP is reasonable.

**Weaknesses:**

1. Less evidance to prove the modality gap existing.
2. It seems this paper is not the first to propose modality gap problems.
3. I doubt that Sharing the parameter space between the vision and language encoders may cause each modality like text or image features not learning well.
4. The object function should consider more about the text modality.

**Questions:**

1. The authors have provided DOSNES projection of the CLIP-encoded image–text pairs from CC3M with ViT-B-32, I wonder how it will be with ViT-L-14 and other image-text datasets?
2.  are the existing CLIP-based methods performing the same phenomen like original CLIP ?
3. The authors have mentioned that there are several works also studying the modality gaps, what is the difference between your findings and these works?
4. Sharing the parameter space between the vision and language encoders may cause each modality like text or image features not learning well, can you provide the parameter comparison between CLIP and AlignCLIP?
5. The proposed object function may cause the learning direction preferring the image based tasks, can you provide more evidance to prove it ?
6. The authors should provide more comarison experiments between existing more recent SOTA methods and AlignCLIP in each task settings.

---

> ### Author Response · Authors · 2024-11-25
> **Response to Reviewer 3Rs9 (part 1/2)**
>
> Thanks a lot for reviewing our paper, recognizing the strengths and providing valuable feedback.
>
> > W1 and W2: Evidence of modality gap
>
> Thanks for the feedback. Since the problem of the modality gap in CLIP has already been extensively introduced in previous work [1, 2], we have provided the corresponding citations in Section 1 and presented the DOSNES visualization, and dedicated the rest of the paper to proposing solutions for mitigating the gap.
>
> > W3: Learning each modality
>
> Thanks for sharing your concerns. However, we sincerely draw your attention to our extensive experimental results on zero-shot image classification, classification with linear probing, classification with natural distribution shift, zero-shot multi-modal retrieval and fine-tuned multi-modal retrieval which **consistently show that SharedCLIP and AlignCLIP achieve superior performance**, therefore, **showing that the image and text features are indeed being learned better**.
>
> > W4. Considering text modality in the objective function.
>
> We have already included this in our submitted paper and **provided the results of using the text-text separation** in the case of intra-modality loss function in Table 8 and 9 of the paper. Our experiments show that the addition of text modality is indeed not helpful and enforcing the separation on the image embeddings outperforms other possibilities.
>
> For more convenience, we report the comparisons in zero-shot image classification here as well. TT stands for text-text separation, II represents image-image separation and II-TT is the case where both text-text and image-image separation have been used.
>
> | **Model**  | **ImageNet1K-top1** | **ImageNet1K-top5** | **CIFAR-100-top1** | **CIFAR-100-top5** | **CIFAR-10-top1** | **CIFAR-10-top5** | **Flowers-102-top1** | **Flowers-102-top5** | **StanfordCars-top1** | **StanfordCars-top5** |
> |------------|---------------------|---------------------|--------------------|--------------------|-------------------|-------------------|----------------------|----------------------|-----------------------|-----------------------|
> | AlignCLIP (TT)  	|  31.1         	|   58.6         	|  31.5         	|  60.9         	|   64.8       	|  95.5        	|  18.7             	| 39.5            	| 10             	|  36.4            	|
> | AlignCLIP (II) | 32.8           	| 60.6           	| 36.5          	| 66.4          	|69.3         	|97.8          	| 18.8            	|40.3             	|11.8               	| 38.1             	|
> | AlignCLIP (II-TT)  |32.4      	|60        	| 31.2       	| 61.8      	|66.4      	|96.9      	| 18.7             	| 39.9             	| 11.8         	|37.1          	|

---

> > ### Author Response · Authors · 2024-11-25
> > **Response to Reviewer 3Rs9 (part 2/2)**
> >
> > In response to your questions:
> >
> > Q1. We respectfully draw your attention to previous work [1, 2], which have extensively studied the existence of the modality gap for various transformer backends such as ViT-L-14.
> >
> > Q2. Yes, we respectfully draw your attention to previous work [1, 2] which have provided these studies.
> >
> > Q3. Previous approach of reducing the gap used naive shifting of the embeddings based on the distance of the centers of each modalities, which could hurt the relative cross-modal distances (shown in Section A.1). As a result, naive way of closing the gap in some experiments improved the performance, and in others, damaged the performance. In our paper, we argued that such shifting can harm the respective cross-modal distances, and show it with an example in Section A.1, Figure 7 and the distances calculated before and after shifting in Eq. 15 and Eq. 16, respectively. Our theoretical argument in Section A.1 shows how the naive shifting approach results in an incorrect retrieval case. Furthermore, our paper firstly finds that increasing the inductive bias towards shared cross-modal encodings by sharing the transformer parameters (i.e., SharedCLIP) is effective in reducing the gap and improving the downstream performances at the same time. Additionally, we find that optimizing an additional semantic-aware intra-modality separation objective function (AlignCLIP) further improves the gap and downstream performance by enforcing a semantically meaningful structure to the embedding space.
> >
> > Q4. Our extensive experiments on zero-shot image classification, classification with linear probing, classification with natural distribution shift, zero-shot multi-modal retrieval and fine-tuned multi-modal retrieval show an **improvement in the results when using ShareCLIP and AlignCLIP**, therefore, **showing that the image and text features are indeed being learned better**.
> >
> > Q5. We respectfully draw your attention to the improvements achieved in **the zero-shot and fine-tuned multi-modal retrieval performances**, showing that our method is not preferring only image-based tasks.
> >
> > Q6. Thanks for your input. We have expanded our SOTA comparisons with the naive shifting approaches utilized in [1, 2] and provide the results in here which shows that AlignCLIP still achieves the best results in the majority of the comparisons. We have also added this comparison to the revised version of our paper.
> >
> > | Model  	| ImageNet1K-top1 | ImageNet1K-top5 | CIFAR-100-top1 | CIFAR-100-top5 | CIFAR-10-top1 | CIFAR-10-top5 | ImageNet-A-top1 | ImageNet-A-top5 | Alignment |
> > |------------|-----------------|-----------------|----------------|----------------|---------------|---------------|-----------------|-----------------|-----------|
> > | CLIP   	| 16.5        	| 34.1        	| 23.7       	| 49.7       	| 51.3      	| 92.7      	| 4           	| 15.8        	| 0.36  	|
> > | CyCLIP 	| 15          	| 32          	| 21.8       	| 48         	| 44.7      	| 85.4      	| 3.3         	| 13.6        	| 0.32  	|
> > | Naive Shifting 	| 16.6          	| 34.6          	| 24.5       	| 52.5         	| 51.2      	| 91.2      	| 43.3         	| 68.8        	| 0.16 (with lambda=0.5) |
> > | SharedCLIP | 17.4        	| 34.9        	| 26         	| 52.1       	| 57        	| 92.9      	| 3.7         	| 14.5        	| 0.45  	|
> > | AlignCLIP  | 17.6        	| 34.9        	| 26.3       	| 54         	| 57.9      	| 93.7      	| 4.2         	| 16.1        	| 0.63  	|
> >
> > We thank you again for your valuable comments and questions. We hope that we have addressed them sufficiently well and hope that you consider raising your scores.
> >
> > **References:**
> >
> > [1] Liang, Victor Weixin, et al. "Mind the gap: Understanding the modality gap in multi-modal contrastive representation learning." Advances in Neural Information Processing Systems 35 (2022): 17612-17625.
> >
> > [2] Schrodi, Simon, et al. "Two Effects, One Trigger: On the Modality Gap, Object Bias, and Information Imbalance in Contrastive Vision-Language Representation Learning." arXiv preprint arXiv:2404.07983 (2024).

---

### Author Response · Authors · 2024-11-25
**Overall Response**

We wish to thank all the reviewers for their constructive feedback and invaluable effort in reviewing our paper. We are pleased to see that the reviewers found our paper excellently well-presented [`3Rs9`, `WiMx`, `mh82`, `6cpP`, `XfLj`, `ZbcG`] and encouraged to see that reviewers [`WiMx`, `6cpP`, `XfLj`] found the motivation of studying and mitigating the modality gap important. The reviewers also recognized the technical novelty and effectiveness of our proposed Intra-Modality Separation objective [`3Rs9`, `XfLj`, `J1QT`, `ZbcG`, `mh82`] and found it beneficial for the community. Furthermore, reviewers viewed our experiments sufficient [`3Rs9`], comprehensive [`WiMx`], broad in scope [`6cpP`, `XfLj`] and appreciated the visualizations as well as the qualitative examples in our manuscript [`6cpP`, `J1QT`], providing clear insights into improvements .

We addressed the concerns raised by the reviewers in the revision and individual responses. Major changes are as follows:

1. **Addition of the ablation study on IMSep loss without sharing parameters**: In response to the concerns about the limitation of AlignCLIP to transformer model and necessity of sharing parameters raised by reviewers \[`WiMx, J1QT, ZbcG`\] and the possibility of using IMSep in two-tower architectures \[`mh82`\], we conducted an ablation study on IMSep loss without sharing parameters using the CC12M dataset and the same hyperparameters reported in the paper. We evaluated the performance of this setting for zero-shot image classification, image classification with linear probing,  image classification with distribution shifts,  zero-shot multi-modal retrieval and  fine-tuned multi-modal retrieval and observed that the **addition of the IMSep loss to the clip loss, without sharing parameters, is indeed effective in reducing the modality gap and enhancing all downstream tasks**. These results suggest that **IMSep loss is also effective in dual-stream two-tower architectures without sharing architecture parameters**.

2. **Enhancement of the SOTA comparison:** Regarding the enhancement of our SOTA comparisons with a more fair experimental setting \[`WiMx`\] and comparing to naive shifting approaches \[`6cpP`, `3Rs9`\], we have fully incorporated reviewers’ suggestions and provided the updated comparisons. Our new comparisons show that the **AlignCLIP model achieves even stronger results** **in comparison to SOTA models**.

3. **Including experiments using additional transformer backends** other than ViT-B-16: Reviewers \[`3Rs9`, `6cpP`, `J1QT`\] have asked whether our method is effective when choosing **other transformer backends**. To study this case, we have **extended** our experiments and repeated them using the **ViT-S-16** backend and provide the results. We observe **the same trend of improvements** when using the ViT-S-16 backend, showing that our method is effective for at least for **two transformer architectures**.

---

> ### Author Response · Authors · 2024-11-25
> **Ablation study of IMSep loss without sharing parameters (part 1/2)**
>
> We present the results of our ablation study, requested by \[`WiMx, J1QT, ZbcG, mh82`\] in this response. In all tables, “IMSep” refers to the case of using the addition of IMSep loss without sharing the transformer between the two modalities.
>
> Starting with the alignment scores, it is observed that **IMSep without sharing parameters improves the alignment score in comparison to CLIP**:
>
> | **Model**  | **CC3M**   | **MSCOCO** | **ImageNet-1K** | **CIFAR-100** | **CIFAR-10** |
> |------------|------------|------------|-----------------|---------------|--------------|
> | CLIP   	| 0.42   	| 0.47   	| 0.41        	| 0.38      	| 0.4      	|
> | SharedCLIP | 0.59   	| 0.62   	| 0.57        	| 0.54      	| 0.54     	|
> | IMSep  	| 0.61   	| 0.64   	| 0.59        	| 0.58      	| 0.6      	|
> | AlignCLIP  | **0.64** | **0.67**   | **0.63**    	| **0.62**  	| **0.64** 	|
>
> Furthermore, the results of the **zero-shot image classification** is as follows. It is observed that IMSep loss without sharing the parameters **achieves an improvement** in the results when compared to the original CLIP model:
>
> | **Model**  | **ImageNet1K-top1** | **ImageNet1K-top5** | **CIFAR-100-top1** | **CIFAR-100-top5** | **CIFAR-10-top1** | **CIFAR-10-top5** | **Flowers-102-top1** | **Flowers-102-top5** | **StanfordCars-top1** | **StanfordCars-top5** |
> |------------|---------------------|---------------------|--------------------|--------------------|-------------------|-------------------|----------------------|----------------------|-----------------------|-----------------------|
> | CLIP   	| 31.4            	| 58.7            	| 28.1           	| 55.9           	| 61.5          	| 95.6          	| 18               	| 39.1             	| 11.6              	| 36.5              	|
> | SharedCLIP | 32.1            	| 59.7            	| 26.4           	| 54.7           	| 56.9          	| 95.2          	| 18.2             	| 38.9             	| 10.7              	| 35.4              	|
> | IMSep  	| 32.1            	| 58.9            	| 31             	| 60.1           	| 61.6          	| 96.2          	| **20.7**         	| **40.5**         	| 9.3               	| 32.7              	|
> | AlignCLIP  | **32.8**        	| **60.6**        	| **36.5**       	| **66.4**       	| **69.3**      	| **97.8**      	| 18.8             	| 40.3             	| **11.8**          	| **38.1**          	|
>
> For **image classification with linear probing**, the following table shows that the IMSep loss without sharing the transformers **substantially improves** the results when compared to the original CLIP model:
> | **Model**  | **ImageNet1K-top1** | **CIFAR-100-top1** | **CIFAR-10-top1** | **Flowers-102-top1** | **StanfordCars-top1** |
> |------------|---------------------|--------------------|-------------------|----------------------|-----------------------|
> | CLIP   	| 50              	| 62.6           	| 85            	| 71.5             	| 42.2              	|
> | SharedCLIP | 51.2            	| 63             	| 85            	| 74.4             	| 40.5              	|
> | IMSep  	| **52.2**              	| 65.9           	| 86.6          	| **79.5**         	| 46.3              	|
> | AlignCLIP  | 51.5        	| **67.4**       	| **87.2**      	| 76.8             	| 45.6          	|
>
> Moreover, we provide the results of **image classification with distribution shifts**. These results also show that the addition of **IMSep improves the accuracy in comparison to the CLIP model**:
> | **Model**  | **ImageNetV2-top1** | **ImageNetV2-top5** | **ImageNet-R-top1** | **ImageNet-R-top5** | **ImageNet-A-top1** | **ImageNet-A-top5** | **ImageNetSketch-top1** | **ImageNetSketch-top5** |
> |------------|---------------------|---------------------|---------------------|---------------------|---------------------|---------------------|-------------------------|-------------------------|
> | CLIP   	| 27.1            	| 53.3            	| 39.8            	| 65.8            	| 6.5             	| 25.4            	| 19.4                	| 41.8                	|
> | SharedCLIP | 27.5            	| 53.5            	| 40.2            	| 67.3            	| 6.7             	| 25.5            	| 20.6                	| **43.2**            	|
> | IMSep  	| 27.5            	| 53.3            	| 40.5            	| **67.5**        	| 6.7             	| 24.5            	| 20.2                	| **43.2**            	|
> | AlignCLIP  | **29.1**        	| **54.4**        	| **41.2**        	| **67.3**        	| **7**           	| **25.6**        	| **20.7**            	| **43.2**            	|

---

> ### Author Response · Authors · 2024-11-25
> **Ablation study of IMSep loss without sharing parameters (part 2/2)**
>
> Additionally, we provide the ablation for the **zero-shot** **multi-modal retrieval** experiments as well. In the following, the performance of image-to-text (I2T) as well as the text-to-image (T2I) retrieval evaluated by R@{1, 5, 10} is reported. As can be seen, the addition of the IMSep loss, without sharing the transformer encoders, **noticeably outperforms the CLIP model**, suggesting the effectiveness of the proposed IMSeploss function regardless of sharing architectures:
>
> | **Model**  | **MSCOCO-I2T-R@1** | **MSCOCO-I2T-R@5** | **MSCOCO-I2T-R@10** | **MSCOCO-T2I-R@1** | **MSCOCO-T2I-R@5** | **MSCOCO-T2I-R@10** | **FLICKR-I2T-R@1** | **FLICKR-I2T-R@5** | **FLICKR-I2T-R@10** | **FLICKR-T2I-R@1** | **FLICKR-T2I-R@5** | **FLICKR-T2I-R@10** |
> |------------|--------------------|--------------------|---------------------|--------------------|--------------------|---------------------|--------------------|--------------------|---------------------|--------------------|--------------------|---------------------|
> | CLIP   	| 31.4           	| 57             	| 68.6            	| 20.5           	| 44.1           	| 55.9            	| 53.2           	| 80.5           	| 88.6            	| 39.9           	| 69             	| 78.5            	|
> | SharedCLIP | 33.5           	| 59.6           	| 70.8            	| 21.8           	| **45.4**       	| **57.3**        	| **58.3**        	| **83.6**       	| **89.8**        	| 42.5           	| 70             	| **79.1**        	|
> | IMSep  	| 33.7           	| **60.8**       	| **71.4**        	| 21.5           	| 45.2           	| 56.8            	| 56.8           	| 82.3           	| 89.2            	| 42             	| 69.9           	| **79.1**        	|
> | AlignCLIP  | **35.1**     	| **60.8**       	| **71.4**        	| **21.9**       	| **45.4**       	| 56.8            	| 57.2           	| 82.3           	| **89.8**        	| **42.7**       	| **70.2**       	| **79.1**        	|
>
> Lastly, the **fine-tuned** **multi-modal retrieval** experiments are provided in the following table:
>
> | **Model**  | **MSCOCO-I2T-R@1** | **MSCOCO-I2T-R@5** | **MSCOCO-I2T-R@10** | **MSCOCO-T2I-R@1** | **MSCOCO-T2I-R@5** | **MSCOCO-T2I-R@10** | **FLICKR-I2T-R@1** | **FLICKR-I2T-R@5** | **FLICKR-I2T-R@10** | **FLICKR-T2I-R@1** | **FLICKR-T2I-R@5** | **FLICKR-T2I-R@10** |
> |------------|--------------------|--------------------|---------------------|--------------------|--------------------|---------------------|--------------------|--------------------|---------------------|--------------------|--------------------|---------------------|
> | CLIP   	| 39.6           	| 67.5           	| 78.3            	| 26.7           	| 53.4           	| 65.7            	| 64.8           	| 87.8           	| 93.9            	| 47.4           	| 75.9           	| 84              	|
> | SharedCLIP | 40.7           	| 69.2           	| 79.6            	| 27.9           	| **55**         	| **66.7**        	| **66.5**       	| **89.1**       	| 94.1            	| 48.9           	| 76.4           	| 84.3            	|
> | IMSep  	| 41             	| **69.3**       	| 79.7            	| 27.8           	| 54.5           	| 66.2            	| 64.8           	| 87.5           	| 93.6            	| 48.5           	| 76.5           	| **84.4**        	|
> | AlignCLIP  | **41.7**       	| **69.3**       	| **80.1**        	| **27.9**       	| 54.8           	| 66.2            	| 66             	| **89.1**       	| **94.5**        	| **49.4**       	| **76.7**       	| **84.4**        	|
>
> This ablation study concludes that the **addition of the IMSep loss to the clip loss is indeed effective in improving the downstream performances**.

---

### Meta-Review · Area_Chair_LcZv · 2024-12-17

**Metareview:**

CLIP has been studied for a long time, but how to effectively handle the pronounced modality gap is still a challenging problem in different downstream tasks. This paper researches this problem from the geometrical point of view, and the authors first visualize this phenomenon using the pre-trained ViT. They propose to reduce the modality gap by sharing the parameter space and highlighting the intra-modality separation capability. For the shared parameters, they build two shared encoders, including a Transformer Encoder and a Projector Layer. For the intra-modal discriminative learning, they formulate an InfoNCE-inspired contrastive loss, which considers visual- and text-level feature similarities combined with Visual-Text feature pairs. Experimental results on classification and retrieval tasks to validate their claims on alignment.

All 7 reviewers pointed out the advantages of the proposed method and they offered a positive rating score, including 6 marginally above the acceptance and 2 acceptance ratings. Generally, the overall novelty of this work is the proposed discriminative criterion based on the cross-modal (diagonal elements of the similarity matrix) information alignment and intra-modal (off-diagonal entries of the similarity matrix) discriminative learning. Hence, the overall idea of this work is good but lower than the excellent bar of 'spotlight', and my final recommendation is accepted with 'poster'.

**Additional Comments On Reviewer Discussion:**

All 7 reviewers would like to accept this paper and the score sheet did not change after rebuttal.

---

### Decision · Program_Chairs · 2025-01-22

Accept (Poster)